# Targets, Mechanisms and Cytotoxicity of Half-Sandwich Ir(III) Complexes Are Modulated by Structural Modifications on the Benzazole Ancillary Ligand

**DOI:** 10.3390/cancers15010107

**Published:** 2022-12-24

**Authors:** M. Isabel Acuña, Ana R. Rubio, Marta Martínez-Alonso, Natalia Busto, Ana María Rodríguez, Nerea Davila-Ferreira, Carl Smythe, Gustavo Espino, Begoña García, Fernando Domínguez

**Affiliations:** 1CIMUS, Universidad de Santiago de Compostela, Avenida Barcelona s/n, 15782 Santiago de Compostela, Spain; 2Departamento de Química, Facultad de Ciencias, Universidad de Burgos, Plaza Misael Bañuelos s/n, 09001 Burgos, Spain; 3Department of Biomedical Science, University of Sheffield, Sheffield S10 2RX, UK; 4Departamento de Ciencias de la Salud, Facultad de Ciencias de la Salud, Universidad de Burgos, Paseo de los Comendadores s/n, 09001 Burgos, Spain; 5Facultad de Ciencias y Tecnologías Químicas-IRICA, University of Castilla-La Mancha, Avda. C. J. Cela 10, 13071 Ciudad Real, Spain

**Keywords:** organometallic iridium complex, DNA binding, mitochondrial damage, proton leak, apoptosis

## Abstract

**Simple Summary:**

Iridium complexes have been reported as potential drug candidates for the treatment of cancer. They are easy to prepare, and their structure allows several modifications that make them extremely versatile as metallodrugs. Thus, the aim of this work was to get a structure–activity relationship study to check the structural versatility of a series of iridium(III) complexes and the way it affects their behavior in both their intrinsic features and inside cells. The most promising complex **1[C,NH]**, showing cyclometallation, targets mitochondria—the cell powerhouse—triggering cell death through mitochondrial membrane depolarization and proton leak. Moreover, in vivo studies in mice confirmed the reduction of the tumor burden. These findings could guide the future design of next generation anticancer drugs able to avoid resistance, as to date, no cancer chemoresistance mechanism can overcome mitochondrial dysfunction.

**Abstract:**

Cancers are driven by multiple genetic mutations but evolve to evade treatments targeting specific mutations. Nonetheless, cancers cannot evade a treatment that targets mitochondria, which are essential for tumor progression. Iridium complexes have shown anticancer properties, but they lack specificity for their intracellular targets, leading to undesirable side effects. Herein we present a systematic study on structure-activity relationships of eight arylbenzazole-based Iridium(III) complexes of type [IrCl(Cp*)], that have revealed the role of each atom of the ancillary ligand in the physical chemistry properties, cytotoxicity and mechanism of biological action. Neutral complexes, especially those bearing phenylbenzimidazole (HL1 and HL2), restrict the binding to DNA and albumin. One of them, complex **1[C,NH-Cl]**, is the most selective one, does not bind DNA, targets exclusively the mitochondria, disturbs the mitochondria membrane permeability inducing proton leak and increases ROS levels, triggering the molecular machinery of regulated cell death. In mice with orthotopic lung tumors, the administration of complex **1[C,NH-Cl]** reduced the tumor burden. Cancers are more vulnerable than normal tissues to a treatment that harnesses mitochondrial dysfunction. Thus, complex **1[C,NH-Cl]** characterization opens the way to the development of new compounds to exploit this vulnerability.

## 1. Introduction

First and second generation Pt(II) metallo-drugs are widely used as anticancer agents in the treatment of testicular, ovarian and colorectal cancers, and DNA is their primary target while hydrolysis of the leaving groups is their activation mechanism. However, the inherent and acquired Pt resistance and severe side toxicity (nephrotoxicity, myelosuppression and neurotoxicity) reduces the range of treatable tumors [1,2,3]. To date, a variety of non-platinum transition metal-based complexes have been intensively exploited for their potential use as anticancer drugs [4,5].

In this context, organometallic complexes have arisen as alternatives to Pt(II) drugs since they have halfway properties between inorganic and organic compounds. In recent years, iridium-based complexes have been widely studied as anticancer derivatives, which exhibit antiproliferative activities against malignant cells [6,7,8]. The factors for the effective design of iridium (III) complexes with antitumoral activity are multiple, such as intrinsic charge of the complex, its lipophilicity and its aqueous solubility. In particular, iridium half-sandwich complexes have aroused considerable interest among the scientific community [9,10,11]. Previous works have studied the structure–activity relationships of iridium(III) families of compounds, modifying the quelating ligands, [12] in combination with mono- [13] and bidentate C^N- [12,14], N^N- [12,14,15,16], N^O- [17], P^O- [18], P^P- [19], C^S-ligands [20] and O^C (carbene) [21] or changing the substituents on Cp* in half-sandwich complexes [17,18].

The presence of the pentamethylcyclopentadienyl ligand (Cp*) in half-sandwich iridium complexes provides stability and hydrophobicity, which improves passive cell uptake. Besides, it promotes the release of the chloride group by increasing the negative charge on the metal, and therefore, it favors aquation and substitution reactions on the metal center, accelerating the coordination to potential biomolecular targets such as nucleobases.

DNA-binding cytotoxic drugs are the first choice in the treatment of many cancers; although the reduction of drug accumulation inside cancer cells [22] increases in the intracellular thiol levels, enhanced DNA repair mechanism [23] and chromatin compaction [24] are important strategies for drug resistance that are difficult to circumvent. Thus, more accessible intracellular targets must be considered. On the other hand, treating tumors based on their unique genetic profile has proved difficult and acquired resistance is common [25]. In this context, mitochondria seems to be a promising target in the design of anticancer drugs since its polarized membrane can retain lipophilic cations and its functionality is altered in cancer cells [26,27,28,29]. Since Otto Warburg first established a link between mitochondria and cancer cells, the role of mitochondria on tumors has been controversial [30,31,32,33]; however, now it is increasingly clear that in proliferating cells, the mitochondrial electron transport chain is needed for the synthesis of aspartate [34,35,36] and pyrimidine [37,38], generated as by-product reactive oxygen species (ROS) [39,40,41]. ROS sensors measure intracellular ROS concentration with a redox-based mechanism and set ROS-specific scavengers’ expression, maintaining the ROS concentration below a toxic threshold, and if the cell cannot do it, triggering apoptosis. Therefore, targeting tumor mitochondria may have various, non-mutually exclusive effects on processes such as (a) mitochondrial ATP production; (b) anabolic pathways required for cell proliferation; (c) permeabilization of the mitochondrial outer membrane, leading to the cytosolic release of pro-apoptotic proteins and (d) an increase in reactive oxygen species (ROS) production by disrupting the transfer of electrons through the electron transport chain (ETC), and thus promoting an oxidative stress response that induce the tumor cell demise. However, the challenge is to be able to proceed selectively. Cancers cannot evade a treatment that disturbs mitochondria, which are essential for tumor progression [32]. Previously, we have found the selectivity of Ir(III) biscyclometallated complexes to act on the mitochondria [42], and recently, Pizarro’s group has described a more potent anticancer compound than cisplatin, ACC25 with the general formula [Ir(η^5^:κ^1^-C_5_Me_4_CH_2_py)(phpy)]PF_6_ (C_5_Me_4_CH_2_py = 2-[(2,3,4,5-tetramethylcyclopentadienyl)methyl] pyridine and phpy=2-phenylpyridine) that is unambiguously localized in the mitochondria [43] as well as another family of nonfluorescent half-sandwich iridium complexes able to generate ROS and disrupt mitochondrial membrane potential [8]. On this basis and considering the wide range of pharmacological properties exhibited by benzimidazole and its derivatives [44,45,46,47], we have prepared four series of neutral and cationic half-sandwich Ir(III)-Cp* complexes bearing eight different arylbenzazole ligands (Figure 1A) according to the notation n[Z,X-Y] (Figure 1B), **1[C,NH-Cl]**, **2[C,NMe-Cl]**, **3[N,NH-Cl]Cl**, **4[N,NMe-Cl]Cl**, **5[CNH_2_,NH-Cl]Cl**, **6[CNH_2_,S-Cl]Cl**, **7[CO,NH-Cl]** and **8[CO,S-Cl]**. Such a diversity has allowed us to modify three different structural parameters, i.e., heteroatom in benzazole (X), the donor atom in the aryl ring (Z), the global charge of the complex and the metallacycle size and geometry (5-membered for series A and B, and 6-membered for series C and D). Moreover, in solution, the chloride leaving group (Y) is replaced with H_2_O or DMSO molecules, depending on their water solubility, leading to aqua or DMSO-adducts. Hence, this study highlights the strong dependence of the chemical and biological behavior of iridium half-sandwich complexes on the nature and position of the atom or group of atoms in the X, Y and Z positions and to prove that small differences in the structure can cause critical changes in their properties. Therefore, this study shows how this class of organometallic anticancer complexes can be fine-tuned to increase their potency without using extended cyclopentadienyl substituents. Overall, structure–activity relationships (SAR) have been established.

## 2. Experimental Section

Materials. The iridium chloride salt (IrCl_3_·nH_2_O) was purchased from Johnson Mattey. The ligands 2-phenylbenzimidazole, 2-(2′-pyridylbenzimidazole), 2-(2′-aminophenylbenzimidazole), 2-(2′-aminophenylbenzothiazole), 2-(2′-hydroxyphenylbenzimidazole) and 2-(2′-hydroxyphenylbenzothiazole) were purchased from Sigma-Aldrich. The ligands 2-phenyl-N-methyl-benzimidazole and 2-(2′-pyridyl)l-N-methyl-benzimidazole were prepared according to slightly modified reported protocols [48]. Deuterated solvents were obtained from SDS and Euriso-top. Unless otherwise specified, all the biological reagents were purchased from Merck Life Science S.L.U., Madrid, Spain. Calf thymus DNA provided as a highly polymerized lyophilized sodium salt was dissolved into doubly-deionized water and sonicated with an MSE-Sonyprep sonicator to reduce the polymer length at approximately 1000 base-pairs. The molar DNA concentration in base pairs, ε (260 nm) = 13,200 M^−1^ cm^−1^ [49], will be expressed as C_P_. BSA was supplied as crystallized and lyophilized powder (≥98%, agarose gel electrophoresis and ≤0.005% fatty acids); the BSA concentration of the stock solutions was spectrophotometrically determined as ε (278 nm) = 45,000 M^−1^ cm^−1^. All aqueous solutions were prepared with MilliQ-grade water using a Direct-Q8UV system from Millipore (Millipore Ibérica S.A., Madrid, Spain). All solvents were of analytical grade. The chemical studies were carried out in aqueous solutions at 25 °C, I = 2.5 mM sodium cacodylate (NaCaC) and pH between 6.0 and 7.1 unless other conditions are stated. All buffers with the biological components were of biological grade and used as received.

Synthesis and characterization.

Synthesis of ligands. Synthesis of 2-Phenyl-N-methyl-benzimidazole (pbim), **[C,NMe]**. In a 250 mL Round-bottomed flask, Cs_2_CO_3_ (3.3509 g, 10 mmol) was added to a solution of 2-phenylbenzimidazole (1 g, 5.15 mmol) in DMSO (30 mL). Some extra DMSO (20 mL) was added to the flask, and the mixture was heated to 80 °C for 1 h 30 min. After that, CH_3_I (481 µL, 7.73 mmol) was added, and the mixture was stirred at RT overnight. Water was added until a white solid was completely precipitated. The solid was filtered off and then dissolved in diethyl ether/toluene to get rid of DMSO traces and evaporate water more easily. The solvent was evaporated to dryness and washed with cold water (2 × 3 mL). The white solid was completely dried in a desiccator until it reached a constant weight. Yield: 548.4 mg (2.63 mmol, 51%).

2-(2′-Pyridyl)l-N-methyl-benzimidazole (pybim), **[N,NMe]**. In a 250 mL round-bottomed flask, Cs_2_CO_3_ (3.3377 g, 10.24 mmol) was added to a solution of 2-(2′-pyridyl)benzimidazole (1 g, 5.12 mmol) in DMSO (25 mL). The mixture was heated to 80 °C for 1 h 30 min. After that, CH_3_I (478 µL, 7.68 mmol) was added, and it was left to RT overnight. Water was added until a white solid was completely precipitated. The solid was filtered off and washed with cold water (2 × 10 mL). The white solid was completely dried in a desiccator until constant weight. Yield: 772.1 mg (3.69 mmol, 72%).

Synthesis of complexes. Synthesis of [(η^5^-Cp*)IrCl(κ^2^-N,C-pbim)], **1[C,NH-Cl]**. In a 100 mL Schlenk flask, the pro-ligand 2-phenylbenzimidazole (151 mg, 0.777 mmol) was added to a solution of [(η^5^-Cp*)IrCl_2_]_2_ (300.5 mg, 0.377 mmol) and sodium acetate (1.02·10^3^ mg, 7.5 mmol) in degassed dichloromethane (20 mL), and the mixture was stirred at room temperature for 20 h under a nitrogen atmosphere. An orange precipitate was formed this time. Water (3 mL) was added to extract the inorganic salts and AcOH compounds, and the mixture was filtered to isolate the crude product. The powder was washed with dichloromethane (1 × 5 mL) and diethyl ether (1 × 5 mL), filtered and dried in a vacuum. The orange solid was dissolved in a mixture of methanol-dichloromethane (18 mL, 2:1) with sodium chloride (6 mg, 0.103 mmol), and the mixture was stirred at room temperature overnight. The solution was concentrated to produce a precipitate. The solid was filtered and washed with dichloromethane (1 × 5 mL), water (1 × 5 mL) and diethyl ether (2 × 5 mL). The resulting orange powder was dried in a vacuum at 50 °C (water bath) for 1 h. The yield was 278.9 mg (0.502 mmol, 65%). The purity (HPLC) was >99%. M_r_ (C_23_H_24_N_2_ClIr) = 556.1287 g/mol. The anal. calcd for C_23_H_24_N_2_ClIr·(CH_2_Cl_2_)_0.4_ was C 47.63, H 4.24 and N 4.75; we found C 47.58, H 4.18 and N 4.37. The ^1^H NMR (400 MHz, DMSO-d_6_, 25 °C) δ 13.60 (s, 1H, H^N-H^), 8.12 (dd, *J* = 5.4, 3.5 Hz, 1H, H^3′^), 7.80 (dd, *J* = 5.6, 3.0 Hz, 1H, H^6′^), 7.70 (dd, *J* = 4.7, 3.8 Hz, 1H, H^c^), 7.51 (d, *J* = 7.3 Hz, 1H, H^f^), 7.46–7.38 (m, 2H, H^e^, H^d^), 7.32 (dd, *J* = 5.7, 3.1 Hz, 2H, H^5′^, H^4′^) and 1.76 (s, 15H, H^Cp*(Me)^) ppm. The ^13^C{^1^H} NMR (101 MHz, DMSO-d_6_, 25 °C) δ 164.2 (s, 1C, C^1′^), 151.8 (s, 1C, C^a^), 139.1 (s, 1C, C^g^), 136.3 (s, 1C, C^6′^), 135.2 (s, 2C, C^2′^, C^b^), 130.9 (s, 1C, C^5′^), 125.0 (s, 1C, C^3′^), 124.5 (s, 1C, C^4′^), 123.6 (s, 1C, C^d^), 123.4 (s, 1C, C^e^), 114.6 (s, 1C, C^c^), 114.3 (s, 1C, C^f^), 95.5 (s, 5C, C^Cp*^) and 8.9 (s, 5C, C^Cp*(Me)^) ppm. NMR spectra were recorded in DMSO-d_6_ due to the low solubility of this product in other solvents, and so the resulting spectra correspond to [(η^5^-Cp*)Ir(DMSO-d_6_)(κ^2^-N,C-pbim)]Cl. FT-IR (ATR, cm^−1^) selected bands: 3423 (w, ν_N-H_), 3139-3111-3064 (m, ν_=CH_), 2962-2911 (m, ν_-CH_), 1592 (s, ν_C=C+C-N_), 1536 (s), 1469-1458 (s, ν_C=N_), 1432 (s), 1379 (w, δ_CH3_), 1278-1262 (s), 1095 (m), 1027 (m), 801 (w, δ_C-C_), 738-729 (vs, δ_CHoop_), 670 (w). MS (FAB+): in DMSO for [(η^5^-Cp*)Ir(DMSO-d_6_)(κ^2^-N,C-Pbim)]Cl: *m*/*z* (%) = 600 (18) ([M-Cl+H]^+^), 522 (100) ([M-Cl-DMSO+H]^+^). For the molar conductivity (H_2_O), no data was available due to low solubility in water or acetonitrile. In terms of solubility, it was soluble in dimethyl sulfoxide (which involves Cl^−^/DMSO replacement) and DMF, partially soluble in methanol and insoluble in water and dichloromethane.

Synthesis of [(η^5^-Cp*)Ir(DMSO)(κ^2^-N,C-pbim)]Cl, **1[C,NH-DMSO]**. In a 100 mL Schlenk flask, DMSO (38 μL, 0.535 mmol) was added to a solution of **1[C,NH-Cl]** (30 mg, 0.054 mmol) in degassed dichloromethane (9 mL), and the mixture was stirred at room temperature for 20 h under a nitrogen atmosphere. The solution was concentrated, and diethyl ether was added to precipitate the product. The solvent was removed with filtration to isolate a white-yellowish powder that was washed with diethyl ether (2 × 5 mL) and dried in a vacuum at 50 °C (water bath) for 1 h. The yield was 25.6 mg (0.040 mmol, 74%). The M_r_ (C_25_H_30_N_2_SOClIr) = 634.2635 g/mol. The anal. calcd for C_25_H_30_N_2_SOClIr·(CH_2_Cl_2_)_0.6_ was C 43.89, H 4.67, N 4.16 and S 4.76; we found C 43.62, H 4.73, N 4.20 and S 4.71. The ^1^H NMR (400 MHz, DMSO-d_6_, 25 °C) δ 14.29 (s, 1H, H^N-H^), 8.00 (dd, *J* = 5.4, 3.5 Hz, 1H, H^3′^), 7.80 (dd, *J* = 5.3, 3.2 Hz, 1H, H^6′^), 7.71 (d, *J* = 8.2 Hz, 1H, H^c^), 7.51 (d, *J* = 6.7 Hz, 1H, H^f^), 7.48–7.40 (m, 2H, H^e^, H^d^), 7.32 (dd, *J* = 5.6, 3.3 Hz, 2H, H^5′^, H^4′^) and 1.77 (s, 15H, H^Cp*(Me)^) ppm. The ^1^H NMR (400 MHz, D_2_O, 25 °C) δ 7.90 (dd, *J* = 8.6, 5.2 Hz, 2H), 7.76–7.69 (m, 1H), 7.61–7.56 (m, 1H), 7.52–7.47 (m, 2H), 7.42–7.36 (m, 2H), 2.94 (s, 3H, H^CH3(DMSO)^), 2.41 (s, 3H, H^CH3(DMSO)^) and 1.80 (s, 15H, H^Cp*(Me)^) ppm. The FT-IR (ATR, cm^−1^) selected bands were 3354 (w, ν_N-H_), 2899 (m, ν_-CH_), 1593 (s, ν_C=C+C-N_), 1540 (s), 1469-1455 (s, ν_C=N_), 1434 (s), 1379 (w, δ_CH3_), 1278 (s), 1117 (vs, ν_S=O_), 1014 (vs), 746 (vs, ν_C-S_) and 685 (w), 427 (vs). The MS (FAB+): *m*/*z* (%) = 600 (4) ([M+H]^+^), 522 (10) ([M-DMSO+H]^+^). It was soluble in dimethyl sulfoxide and water and partially soluble in methanol. 

Synthesis of [(η^5^-Cp*)IrCl(κ^2^-N,C-NMepbim)], **2[C,NMe-Cl]**. In a 100 mL Schlenk flask, the pro-ligand 2-phenyl-N-Methylbenzimidazole (54 mg, 0.0.259 mmol) was added to a solution of [(η^5^-Cp*)IrCl_2_]_2_ (100 mg, 0.129 mmol) and sodium acetate (360 mg, 2.646 mmol) in degassed dichloromethane (18 mL), and the mixture was stirred at room temperature for 20 h under a nitrogen atmosphere. Hexane was added to precipitate the complex, and the solution was concentrated. The orange solid was filtered and then dissolved in a mixture of methanol-dichloromethane (13 mL, 5:8) with sodium chloride (13 mg, 0.224 mmol), and the mixture was stirred at room temperature overnight. The solvent was evaporated to dryness and washed with water (2 × 5 mL) and diethyl ether (2 × 5 mL). The resulting orange powder was dried in a vacuum at 50 °C (water bath) for 1 h. The yield was 75.4 mg (0.132 mmol, 51%). The purity (HPLC) was >99%. The M_r_ (C_24_H_26_N_2_ClIr) = 570.1555 g/mol. The anal. calcd for C_24_H_26_N_2_ClIr was C 50.56, H 4.60 and N 4.91; we found C 50.51, H 4.56 and N 5.18. The ^1^H NMR (400 MHz, DMSO-d_6_, 25 °C) δ 8.18–8.12 (m, 1H, H^3′^), 7.95 (m, 1H, H^c^), 7.89–7.83 (m, 1H, H^6′^), 7.56–7.49 (m, 3H, H^d^, H^e^, H^f^), 7.39–7.32 (m, 2H, H^4′^, H^5′^), 4.35 (s, 3H, H^h^) and 1.75 (s, 15H, H^Cp*(Me)^). The ^13^C{^1^H} NMR (101 MHz, DMSO-d_6_, 25 °C) δ 162.0 (s), 153.7 (s), 137.8 (s), 136.9 (s), 136.1 (s), 134.5 (s), 131.1 (s), 126.5 (s), 124.6 (s), 124.3 (s), 124.2 (s), 115.3 (s), 112.5 (s), 95.8 (s, 5C, C^Cp*^), 32.4 (s, 1C, C^h^) and 8.9 (s, 5C, C^Cp*(Me)^) ppm. The FT-IR (KBr, cm^−1^) selected bands were 2904 (w, ν_-CH_), 1579 (s, ν_C=C + C-N_), 1506 (s), 1479-1456 (s, ν_C=N_), 1420-1402 (s), 1376 (m, δ_CH3_), 1276 (m), 1153 (m), 1027-1010 (s), 828 (m, δ_C-C_), 764 (vs), 727 (vs, δ_CHoop_), 612 (s) and 449 (s). The MS (ESI+) in methanol was *m*/*z* (%) = 565 (79) ([M-Cl+OCH_3_-H]^+^), 535 (100) ([M-Cl]^+^). The molar conductivity (CH_3_CN) was 13.0 S·cm^2^·mol^−1^. It was soluble in dimethyl sulfoxide (with substitution of the Cl^−^), partially soluble in methanol and acetonitrile and insoluble in water and dichloromethane.

Synthesis of [(η^5^-Cp*)IrCl(κ^2^-N,N-pybim)]Cl, **3[N,NH-Cl]**. In a 100 mL Schlenk flask, the ligand pybim (0.0505 g, 0.259 mmol) was added to a solution of [IrCl_2_(Cp*)]_2_ (0.1002 g, 0.126 mmol) in dichloromethane (14 mL), and the mixture was stirred at room temperature for 20 h and under a nitrogen atmosphere. The solution was concentrated, and the product was precipitated with hexane and filtered off. The resulting yellow powder was dried in a vacuum at 50 °C (water bath) for 1 h. The yield was 147.7 mg (0.249 mmol, 98%). The purity (HPLC) was >98%. The M_r_ (C_22_H_24_N_3_Cl_2_Ir) = 593.5771 g/mol. The anal. calcd for C_22_H_24_N_3_Cl_2_Ir·(CH_2_Cl_2_)_1.1_ was C 40.50, H 3.78 and N 5.45; we found C 40.39, H 3.84 and N 6.12. The ^1^H NMR (400 MHz, CDCl_3_, 25 °C) δ 16.21 (s, 1H, H^N-H^), 9.67 (d, *J* = 8.2 Hz, 1H, H^3′^), 8.76 (d, *J* = 5.7 Hz, 1H, H^6′^), 8.15 (t, *J* = 7.8 Hz, 1H, H^4′^), 8.01 (d, *J* = 8.2 Hz, 1H, H^c^), 7.65 (d, *J* = 8.1 Hz, 1H, H^f^), 7.61–7.53 (m, 1H, H^5′^), 7.53–7.45 (m, 1H, H^d^), 7.42 (t, *J* = 7.7 Hz, 1H, H^e^) and 1.75 (s, 15H, H^Cp(Me)^) ppm. The ^13^C{^1^H} NMR (101 MHz, CDCl_3_, 25 °C) δ 152.3 (s, 1C, C^a^), 150.64 (s, 1C, C^6′^), 148.2 (s, 1C, C^2′^), 140.8 (s, 1C, C^4′^), 138.7 (s, 1C, C^g^), 135.5 (s, 1C, C^b^), 127.4 (s, 1C, C^5′^), 126.5 (s, 1C, C^3′^), 126.2 (s, 1C, C^d^), 125.1 (s, 1C, C^e^), 116.2 (s, 1C, C^d^), 115.8 (s, 1C, C^f^), 88.3 (s, 1C, C^CpC^) and 9.7 (s, 1C, C^Cp(Me)^) ppm. The FT-IR (ATR, cm^−1^) selected bands were 3389 (vs, ν_N-H_), 3029 (w, ν _C=CH_), 2963-2915 (w, ν_-CH_), 1612-1594 (m, ν_C=C + C-N_), 1484-1457-1447 (vs, ν_C=N_), 1382 (m, δ_CH3_), 1325 (m), 1261 (m), 1028 (s), 795 (m, δ_C-C_) and 760 (s, δ_CHoop_). The MS (FAB+) was *m*/*z* (%) = 648 (4), 559 (12) ([M-Cl+H]^+^), 522 (5) ([M-2Cl-H]^+^). The molar conductivity (CH_3_CN) was 27.1 S·cm^2^·mol^−1^. It was soluble in water, dichloromethane, chloroform, acetonitrile and acetone. 

Synthesis of [(η^5^-Cp*)IrCl(κ^2^-N,N-pyMebim)]Cl, **4[N,NMe-Cl]**. The synthesis was performed as for **3[N,NH-Cl]** in the presence of the ligand 2-(2′-pyridyl)-N-methylbenzimidazole (0.0344 g, 0.164 mmol) and [IrCl_2_(Cp*)]_2_ (0.0574 g, 0.074 mmol) dichloromethane (6 mL). The product was precipitated with diethyl ether as a yellow powder. The yield was 81.8 mg (0.135 mmol, 91%). The purity (HPLC) was >97%. The M_r_ (C_23_H_26_N_3_Cl_2_Ir) = 607.6039 g/mol. The anal. calcd for C_23_H_26_N_3_Cl_2_Ir·(CH_2_Cl_2_)_0.5_(H_2_O)_0.4_ was C 42.95, H 4.26 and N 6.39; we found C 42.99, H 4.47 and N 5.96. The ^1^H NMR (400 MHz, CDCl_3_, 25 °C) δ 9.29 (d, *J* = 8.0 Hz, 1H, H^3′^), 8.89 (d, *J* = 5.6 Hz, 1H, H^6′^), 8.41 (t, *J* = 8.0 Hz, 1H, H^4′^), 7.74–7.63 (m, 3H, H^f^, H^5′^, H^c^), 7.62–7.55 (m, 1H, H^d^), 7.53–7.48 (m, 1H, H^e^), 4.70 (s, 3H, H^Me^) and 1.78 (s, 15H, H^Cp(Me)^) ppm. The ^13^C{^1^H} NMR (101 MHz, CDCl_3_, 25 °C) δ 152.0 (s, 1C, C^6′^), 151.8 (s, 1C, C^a^), 146.9 (s, 1C, C^2′^), 141.6 (s, 1C, C^4′^), 138.1 (s, 1C, C^g^), 136.9 (s, 1C, C^b^), 128.1 (s, 1C, C^5′^), 127.1 (s, 1C, C^3′^), 126.6 (s, 1C, C^d^), 125.7 (s, 1C, C^e^), 116.9 (s, 1C, C^f^), 112.7 (s, 1C, C^c^), 88.9 (s, 1C, C^CpC^) and 9.8 (s, 1C, C^Cp(Me)^) ppm. The FT-IR (ATR, cm^−1^) selected bands were 3444-3379 (m, ν_N-H_), 3069 (w, ν _C=CH_), 1606 (m, ν_C=C + C-N_), 1524 (m), 1490-1468-1441 (vs, ν_C=N_), 1354-1334 (w, δ_CH3_), 1153 (m), 1030 (s), 832 (w), 792 (m, δ_C-C_), 756-742 (vs, δ_CHoop_), 582 (m), 545 (m) and 507 (w). The MS (FAB+) was *m*/*z* (%) = 572 (33) ([M-Cl]^+^), 536 (4) ([M-2Cl-H]^+^) and 363 (5) ([M-Cl-NMepybzIm-H]^+^). The molar conductivity (CH_3_CN) was 140.9 S·cm^2^·mol^−1^. It was soluble in water, methanol, dichloromethane and chloroform.

Synthesis of [(η^5^-Cp*)IrCl(κ^2^-N,N-apbim)]Cl, **5[CNH_2_,NH-Cl]**. The synthesis was performed as for **4[N,NMe-Cl]** in the presence of the ligand 2-(2′-aminophenyl)benzimidazole (0.0546 g, 0.261 mmol) and [IrCl_2_(Cp*)]_2_ (0.1000 g, 0.126 mmol) dichloromethane (14 mL). The precipitate was a yellow powder. The yield was 122.0 mg (0.201 mmol, 80%). The purity (HPLC) was >97%. The M_r_ (C_23_H_26_N_3_Cl_2_Ir) = 607.6039 g/mol. The anal. calcd for C_23_H_26_N_3_Cl_2_Ir·(CH_2_Cl_2_)_0.3_ was C 44.25, H 4.18 and N 6.13; we found C 44.21, H 4.24 and N 6.62. The ^1^H NMR (400 MHz, DMSO-d_6_, 25 °C) δ 14.61 (s, 1H, H^N-H^), 8.23 (d, *J* = 9.7 Hz, 1H, H^NH2^), 8.08 (d, *J* = 7.6 Hz, 1H, H^3′^), 7.85 (d, *J* = 11.0 Hz, 1H, H^NH2^), 7.75–7.70 (m, 1H, H^c^), 7.67 (dd, *J* = 5.6, 3.7 Hz, 1H, H^f^), 7.63 (t, *J* = 7.7 Hz, 1H, H^5′^), 7.52 (d, *J* = 7.8 Hz, 1H, H^6′^), 7.47–7.36 3m, 3H, H^4′^, H^d^, H^e^) and 1.46 (d, *J* = 63.7 Hz, 15H, H^Cp(Me)^) ppm. The ^13^C{^1^H} NMR (101 MHz, DMSO-d_6_, 25 °C) δ 147.2 (s, 1C, C^a^), 141.9 (s, 1C, C^1′^), 139.4 (s, 1C, C^g^), 134.0 (s, 1C, C^b^), 131.5 (s, 1C, C^5′^), 129.0 (s, 1C, C^3′^), 126.0 (s, 1C, C^4′^), 124.5 (s, 1C, C^d^), 123.4 (s, 1C, C^e^), 121.6 (s, 1C, C^2′^), 121.0 (s, 1C, C^6′^), 118.8 (s, 1C, C^c^ or C^f^), 112.6 (s, 1C, C^f^ or C^5′^), 86.0 (s, 1C, C^CpC^) and 8.1 (s, 1C, C^Cp(Me)^) ppm. The FT-IR (ATR, cm^−1^) selected bands were 3425 (w, ν_O-H_), 3025 (vs, ν_=CH,_ ν_NH2_), 2962-2900 (vs, ν_NH2+(hydrogen bonds)_), 1620-1598 (w, ν_C=C + C-N_), 1541 (m), 1485 (s), 1463-1450 (vs, ν_C=N_), 1417 (s), 1382 (m, δ_CH3_), 1325 (m), 1161 (m), 1032 (m), 799 (m, δ_C-C_) and 762-747 (s, δ_CHoop_). The MS (FAB+) was *m*/*z* (%) = 573 (25) ([M-Cl+H]^+^), 537 (21) ([M-2Cl]^+^). The molar conductivity (CH_3_CN) was 50.6 S·cm^2^·mol^−1^. It was soluble in water, methanol, dimethyl sulfoxide and acetonitrile. It was insoluble in dichloromethane and acetone. 

Synthesis of [(η^5^-Cp*)IrCl(κ^2^-N,N-apbtz)]Cl, **6[CNH_2_,S-Cl]**. The synthesis was performed as for **4[N,NMe-Cl]** in the presence of the ligand 2-(2′-aminophenyl)benzothiazole (0.0546 g, 0.258 mmol) and [IrCl_2_(Cp*)]_2_ (0.0999 g, 0.125 mmol) dichloromethane (14 mL). The precipitate was a yellow powder. The yield was 114.9 mg (0.184 mmol, 73%). The purity (HPLC) was >98%. The M_r_ (C_23_H_25_N_2_SCl_2_Ir) = 624.6553 g/mol. The anal. calcd for C_23_H_25_N_2_SCl_2_Ir·(CH_2_Cl_2_)_1.1_ was C 40.31, H 3.82, N 3.90 and S 4.47; we found C 40.19, H 3.92, N 3.55 and S 4.47. The ^1^H NMR (400 MHz, CDCl_3_, 25 °C) δ 10.58 (d, *J* = 10.9 Hz, 1H, H^NH2^), 8.77 (d, *J* = 7.8 Hz, 1H, H^6′^), 8.34 (d, *J* = 8.3 Hz, 1H, H^f^), 7.97 (d, *J* = 7.9 Hz, 1H, H^c^), 7.82 (d, *J* = 7.7 Hz, 1H, H^3′^), 7.68 (t, *J* = 7.8 Hz, 1H, H^5′^), 7.62 (t, *J* = 7.1 Hz, 1H, H^e^), 7.56 (t, *J* = 7.4 Hz, 1H, H^d^), 7.38 (t, *J* = 7.5 Hz, 1H, H^4′^), 5.87 (d, *J* = 10.5 Hz, 1H, H^NH2^) and 1.54 (s, 15H, H^Cp(Me)^) ppm. The ^13^C{^1^H} NMR (101 MHz, CDCl_3_, 25 °C) δ 165.4 (s, 1C, C^a^), 150.3 (s, 1C, C^g^), 140.4 (s, 1C, C^1′^), 134.0 (s, 1C, C^5′^), 132.5 (s, 1C, C^b^), 130.3 (s, 1C, C^3′^), 128.1 (s, 1C, C^e^), 127.8 (s, 1C, C^4′^), 127.7 (s, 1C, C^d^), 125.7 (s, 1C, C^f^), 125.4 (s, 1C, C^2′^), 124.8 (s, 1C, C^6′^), 122.6 (s, 1C, C^c^), 87.8 (s, 1C, C^CpC^) and 9.3 (s, 1C, C^Cp(Me)^) ppm. The FT-IR (ATR, cm^−1^) selected bands were 3408 (w, ν_O-H_), 3024 (m, ν_=CH,_ ν_NH2_), 2963-2922 (s, ν_NH2+(hydrogen bonds)_), 1607 (w, ν_C=C + C-N_), 1574 (w), 1477-1449 (vs, ν_C=N_), 1430 (s), 1380 (m, δ_CH3_), 1322 (w), 1254 (m), 1176 (m), 1078 (m, ν_C=S_), 1031-997 (s), 789 (vs, δ_C-C_), 755 (vs, δ_CHoop_), 718 (s) and 689 (m). The MS (FAB+) was *m*/*z* (%) = 590 (16) ([M-Cl+H]^+^), 554 (9) ([M-2Cl]^+^). The molar conductivity (CH_3_CN) was 79.6 S·cm^2^·mol^−1^. It was soluble in dichloromethane, chloroform, acetonitrile and dimethyl sulfoxide. 

Synthesis of [(η^5^-Cp*)IrCl(κ^2^-O,N-hpbim)], **7[CO,NH-Cl]**. In a 100 mL Schlenk flask, the ligand 2-(2′-hydroxyphenyl)benzimidazole (0.0541 g, 0.257 mmol) was added to a solution of [IrCl_2_(Cp*)]_2_ (0. 0999 g, 0.125 mmol) and Et_3_N (37 μL, 0.266 mmol) in dichloromethane (15 mL), and the mixture was stirred at room temperature for 20 h and under a nitrogen atmosphere. The product was filtered and washed with water and diethyl ether. The resulting yellow powder was dried in a vacuum at 50 °C (water bath) for 1 h. The yield was 116.8 mg (0.204 mmol, 81%). The purity (HPLC) was >95%. The M_r_ (C_23_H_24_N_2_OClIr) = 572.1281 g/mol. The anal. calcd for C_23_H_24_N_2_OClIr·(CH_2_Cl_2_)_0.3_ was C 46.83, H 4.15 and N 4.69; we found C 46.89, H 4.13 and N 4.28. The ^1^H NMR (400 MHz, DMSO-d_6_/MeOD-d_4_, 25 °C) δ 7.78 (dd, *J* = 7.9, 1.7 Hz, 1H, H^3′^), 7.66 (dd, *J* = 7.0, 1.0 Hz, 1H, H^c^), 7.46 (td, *J* = 7.5, 1.2 Hz, 1H, H^e^), 7.41 (td, *J* = 7.5, 1.2 Hz, 1H, H^d^), 7.34 (dd, *J* = 7.2, 1.0 Hz, 1H, H^f^), 7.28 (t, *J* = 7.6, 1.0 Hz, 1H, H^5′^), 6.92 (dd, *J* = 8.4, 1.1 Hz, 1H, H^6′^), 6.75 (td, *J* = 7.6, 1.0 Hz, 1H, H^4′^) and 1.47 (s, 15H, H^Cp(Me)^) ppm. The ^13^C{^1^H} NMR (101 MHz, DMSO-d_6_/MeOD-d_4_, 25 °C) δ 166.0 (s, 1C, C^1′^), 148.6 (s, 1C, C^a^), 140.3 (s, 1C, C^g^), 135.1 (s, 1C, C^b^), 133.4 (s, 1C, C^5′^), 128.3 (s, 1C, C^3′^), 125.8 (s, 1C, C^d^), 124.8 (s, 1C, C^e^), 123.9 (s, 1C, C^6′^), 118.5 (s, 1C, C^4′^), 117.6 (s, 1C, C^f^), 116.1 (s, 1C, C^2′^), 113.6 (s, 1C, C^c^), 93.1 (s, 1C, C^CpC^) and 8.5 (s, 1C, C^Cp(Me)^) ppm. The FT-IR (ATR, cm^−1^) selected bands were 3165-3141 (m), 3044 (w, ν_=CH_), 2983 (w, ν_-CH_), 1620 (m, ν_C-N_), 1600 (s, ν_C=C_), 1552-1532 (m), 1476-1444 (vs, ν_C=N_), 1315 (s), 1259 (s, ν_C-O_), 1137 (m), 1033 (m), 858 (m), 771-749 (s, δ_CHoop_) and 689 (w). The MS (FAB+) (for the derivative obtained after Cl^−^/DMSO substitution) was *m*/*z* (%) = 616 (10) ([M-Cl+DMSO+H]^+^), 538 (72) ([M-DMSO+H]^+^). It was soluble in a mixture of dimethyl sulfoxide/methanol (3:2). It was slightly soluble in methanol. It was insoluble in water, dimethyl sulfoxide, dichloromethane, acetonitrile and acetone.

Synthesis of [(η^5^-Cp*)IrCl(κ^2^-O,N-hpbtz)], **8[CO,S-Cl]**. The synthesis was performed as for **7[CO,NH-Cl]** in the presence of the ligand 2-(2′-hidroxyphenyl)benzothiazole (0.0541 g, 0.259 mmol), [IrCl_2_(Cp*)]_2_ (0. 0999 g, 0.125 mmol) and Et_3_N (37 μL, 0.266 mmol) in dichloromethane (12 mL). The precipitate was a yellow powder. The yield was 81.4 mg (0.138 mmol, 55%). The purity (HPLC) was >99%. The M_r_ (C_23_H_23_NOSClIr) = 589.1795 g/mol. The anal. calcd for C_23_H_23_NOSClIr·(CH_2_Cl_2_)_0.3_ was C 45.53, H 3.87, N 2.28 and S 5.22; we found C 45.51, H 4.00, N 2.01 and S 5.06. The ^1^H NMR (400 MHz, CDCl_3_, 25 °C) δ 8.28 (dd, *J* = 8.4, 0.6 Hz, 1H, H^f^), 7.77 (dd, *J* = 8.0, 0.6 Hz, 1H, H^c^), 7.54–7.49 (m, 2H, H^3′,e^), 7.37 (td, *J* = 7.3, 1.1 Hz, 1H, H^d^), 7.31–7.23 (m, 1H, H^5′^), 7.06–7.01 (m, 1H, H^6′^), 6.56 (ddt, *J* = 8.1, 7.0, 1.1 Hz, 1H, H^4′^) and 1.43 (s, 15H, H^Cp(Me)^) ppm. The ^13^C{^1^H} NMR (101 MHz, CDCl_3_, 25 °C) δ 167.7 (s, 1C, C^1′^), 165.4 (s, 1C, C^a^), 151.2 (s, 1C, C^g^), 133.6 (s, 1C, C^5′^), 131.9 (s, 1C, C^b^), 129.2 (s, 1C, C^3′^), 126.9 (s, 1C, C^e^), 125.7 (s, 1C, C^d^), 125.3 (s, 1C, C^f^), 124.4 (s, 1C, C^6′^), 121.7 (s, 1C, C^2′^), 121.4 (s, 1C, C^c^), 116.5 (s, 1C, C^4′^), 84.6 (s, 1C, C^CpC^) and 9.2 (s, 1C, C^Cp(Me)^) ppm. The FT-IR (ATR, cm^−1^) selected bands were 3408 (w, ν_O-H_), 3050 (w, ν_=CH_), 2964 (w, ν_-CH_), 1599 (s, ν_C=C+C-N_), 1543 (m), 1492-1453 (vs, ν_C=N_), 1377 (m, δ_CH3_), 1325 (s), 1240-1223-1209 (s, ν_C-O_), 1147 (m), 1077 (w, ν_C=S_), 1032 (m), 771-733 (s, δ_CHoop_) and 689 (w). The MS (FAB+) was *m*/*z* (%) = 590 (23) ([M]^+^), 555 (100) ([M-Cl+H]^+^). The molar conductivity (CH_3_CN) was 10.5 S·cm^2^·mol^−1^. It was soluble in dichloromethane, chloroform, acetonitrile and acetone. It was slightly soluble in methanol and insoluble in water.

Synthetic methods. All synthetic manipulations were carried out under an atmosphere of dry, oxygen-free nitrogen using standard Schlenk techniques. The solvents were dried and distilled under a nitrogen atmosphere before use. The elemental analyses were performed in a PerkinElmer 2400 CHN microanalyzer. The analytical data for the new compounds were obtained from crystalline samples when possible. In some cases, the data was reasonably accurate, but in others, the agreement between calculated and found values for carbon was >0.4% so that solvent molecules were introduced in the molecular formulas to improve agreement. The IR spectra were recorded on a Jasco FT/IR-4200 spectrophotometer (4000–400/cm range) with a Single Reflection ATR Measuring Attachment. The FAB^+^ mass spectra (position of the peaks in DA) were recorded with an Autospec spectrometer. The isotopic distribution of iridium matched very closely with the calculated values for the expected complex cation in every case. The NMR samples were prepared under a nitrogen atmosphere by dissolving a suitable amount of the compound in 0.5 mL of the respective oxygen-free deuterated solvent, and the spectra were recorded at 298 K (unless otherwise stated) on a Varian Unity Inova-400 (399.94 MHz for ^1^H; 161.9 MHz for ^31^P; 100.6 MHz for ^13^C). Typically, 1D ^1^H NMR spectra were acquired with 32 scans into 32 k data points over a spectral width of 16 ppm. The ^1^H and ^13^C{^1^H} chemical shifts were internally referenced to TMS via 1,4-dioxane in D_2_O (δ = 3.75 and 67.19 ppm, respectively) or via the residual ^1^H and ^13^C signals of the corresponding solvents, CDCl_3_ (δ 7.26 and 77.16 ppm) and (CD_3_)_2_SO (δ 2.50 and 39.52 ppm), according to the values reported by Fulmer et al [50]. Chemical shift values are reported in ppm and coupling constants (J) in hertz. The splitting of proton resonances in the reported ^1^H NMR data was defined as s = singlet, d = doublet, t = triplet, st = pseudotriplet, q = quartet, sept = septet, m = multiplet and bs = broad singlet. The 2D NMR spectra such as ^1^H−^1^H gCOSY, ^1^H−^1^H NOESY, ^1^H−^13^C gHSQC and ^1^H−^13^C gHMBC were recorded using standard pulse sequences. The probe temperature (±1 K) was controlled by a standard unit calibrated with methanol as a reference. All NMR data processing was carried out using MestReNova version 10.0.2.

HPLC Purity measurements by HPLC were performed on a Waters instrument (Alliance module e2695), using a Mediterranean Sea C18 column, 250 × 4.6 mm, 5 µm. The mobile phase used was MilliQ-water (0.1% TFA, solvent A) and HPLC grade acetonitrile (0.1% TFA, solvent B). Gradient elution was used following the next method: 0–3 min, isocratic 95% A (5% B); 3–17 min, linear gradient from 95% A (5% B) to 0% A (100% B); 17–23, isocratic 0% A (100% B). A PDA detector was used for peak detection monitoring at 254, 350, 450 and 550 nm channels. All peaks were manually integrated at 254 nm to obtain the percentage area. Samples were prepared by dissolving some solid in DMSO. Then, acetonitrile and MilliQ-water were qualitatively added while avoiding precipitation. The solutions were filtered with hydrophilic PVDF syringe filters (Filter-Lab, Barcelona, Spain) before injections. Each sample was analysed at least twice (injection volume 20 µL) to ensure reproducibility. Complexes **1[C,NH]** and **2[C,NMe]** show two peaks corresponding to the equilibrium between the Ir-DMSO and the Ir-acetonitrile species since both solvents exhibit coordinating abilities [51] (see Appendix A). In these cases, both areas were added together for purity measurements. The chromatogram of DMSO was also recorded (Appendix A).

Solubility. The solubility of the complexes was qualitatively assessed. Different solvents were added drop-by-drop while shaking to a small amount of complex (around 1–2 mg) in a vial until the solvent added exceeded 1 mL. Afterwards, it was checked if the solid was or not completely dissolved and/or if the solution got coloured.

The pH measurements were carried out with a Metrohm 16 DMS Titrino pH meter fitted out with a combined glass electrode with a 3 M KCl solution as a liquid junction. HClO_4_ and NaOH solution were used to adjust the pH.

Aquation/solvolysis. Chloride displacement was qualitatively monitored by ^1^H-NMR for some of the complexes in DMSO-d_6_. The replacement of the chloride by a DMSO molecule was observed as some peaks shifted and some new peaks appeared corresponding to the methyl groups of the DMSO molecule. Aquation/solvolysis equilibria were studied by kinetic UV-vis measurements.

The UV-vis measurements were performed with a Hewlett-Packard 8453A (Agilent Technologies, Palo Alto, CA, USA) photodiode array spectrophotometer with a Peltier temperature control system. Titrations were carried out at 25 °C by adding increasing amounts of DNA solution to the iridium complex solution. The pK_a_ values were measured by analysing the absorbance change of iridium complex solutions recorded at different pH values. The melting experiments were carried out using a DNA solution varying the C_Ir-complex_/C_DNA_ (C_D_/C_P_) ratio from 0 to 1, keeping C_P_ constant and recording the absorbance spectra while heating from 25 to 90 °C at 0.3 °C/min.

The slow kinetic measurements were performed with the previous spectrophotometer while the fast kinetic measurements were performed in a stopped flow Bio-Logic SFM-300 spectrophotometer with an absorbance detection system and dead time below 3 ms. The kinetic curves, obtained averaging out at least five shots, were analysed with the Jandel (AISN software, Mapleton, OR) fitting package.

Differential scanning calorimetry (DSC) studies were performed in a nano DSC (TA Instruments, Newcastle, USA) to determine the melting temperature of DNA in the absence and in the presence of different amount of the Ir(III) complexes. Cells were 300 μL platinum capillary tubes. Measurements were performed by heating the dye/polynucleotide system from 20 to 90 °C, at 1 °C/min scan rate and 3 atm pressure. To reduce the formation of bubbles upon heating to a minimum, the reference and the sample solutions were previously degassed for 30 min in a degassing station (TA Instruments, Newcastle, USA). The thermograms recorded were analysed with the NanoAnalyze 2.0 software. The buffer–buffer baseline was run at least five heating/cooling cycles until the heating was reproducible, and then, it was subtracted from the sample data.

Circular dichroism (CD) measurements were performed with a MOS-450 biological spectrophotometer (Bio-Logic SAS, Claix, France) fitted out with 1.0 cm path length cells. Titrations were carried out at 25 °C by adding increasing amounts of iridium complex solution to the DNA solution. Spectrograms were obtained in the 200−700 nm range at 2 nm/s speed. Molar ellipticity (Deg M^−1^ cm^−1^) was calculated using [θ] = 100·θ/C_P_·l, where C_P_ is the polynucleotide concentration and l is the cell light path (cm).

Viscosity measurements were performed using a Micro-Ubbelohde viscometer whose temperature was controlled externally (25 ± 0.1 °C). The flow time was measured with a digital stopwatch. Mean values of triplicated measurements were taken to evaluate the DNA viscosity in the absence (η_0_) and in the presence, (η) of iridium complexes. The viscosity readings were expressed as L/L_0_ = (η/η_0_)^1/3^ versus the C_D_/C_P_ ratio. The data was analysed using L/L_0_ = (t − t_0_)/(t_DNA_ − t_0_) where t_0_ and t_DNA_ are the buffer and polynucleotide solution flow times, respectively, whereas t is the flow time of the Ir-complex/DNA mixture after the stabilization period.

Agarose Gel electrophoresis of plasmid DNA Overnight cultures of the strain DH5α of *Escherichia coli* were isolated with the Quiagen DNA purification kit. Then, 20 µM (in base pairs) of plasmid DNA (pUC18) was incubated with different concentrations of the synthesized iridium complexes overnight at 37 °C. Afterwards, 2 μL of loading buffer were added to 10 μL of the incubated samples and loaded onto a 1% agarose gel. Electrophoresis was run in TAE 1× at 6.5 V/cm during 2.5 h. Then, the agarose gel was stained with ethidium bromide 1 μg/mL and visualized with a Gel Doc XR+ Imaging System (Bio-Rad, Hercules, CA, USA).

X-ray Crystallography. Single crystals of **1[C,NH-Cl]**, **3[N,NH-Cl]·2H_2_O**, **4[N,NMe-Cl]·H_2_O, 5[CNH_2_,NH]·H_2_O** and **8[CO,S-Cl]** were mounted on a glass fiber and transferred to a Bruker X8 APEX II CCD-based diffractometer equipped with a graphite-monochromated Mo Kα radiation source (λ = 0.71073 Å). The highly redundant data sets were integrated using SAINT [47] and corrected for Lorentz and polarization effects. The absorption correction was based on the function fitting to the empirical transmission surface as sampled by multiple equivalent measurements with the program SADABS [52]. The software package SHELXTL (version 6.10) [53] was used for space group determination, structure solution and refinement with full- matrix least-squares methods based on F2. A successful solution by direct methods provided most non-hydrogen atoms from the E map. The remaining non-hydrogen atoms were located in an alternating series of least-squares cycles and difference Fourier maps. All non-hydrogen atoms were refined with anisotropic displacement coefficients. Hydrogen atoms were placed using a “riding model” and included in the refinement at calculated positions. The CCDC reference numbers for **1[C,NH-Cl]**, **3[N,NH-Cl]·2H_2_O**, **4[N,NMe-Cl]·H_2_O, 5[CNH_2_,NH]·H_2_O** and **8[CO,S-Cl]** are 2082270, 2082271, 2082272, 2082273 and 2082274, respectively. The thermal ellipsoid molecular structures were plotted using the ORTEP-III program for crystal structure illustrations.

Cell culture. Epithelial ovarian carcinoma cells (A2780) that acquired resistance to cisplatin epithelial ovarian carcinoma cells (A2780cis), colon adenocarcinoma cells (SW480) and epithelial lung adenocarcinoma cells (A549) were provided by Sigma Aldrich. A549 and SW480 cells were grown in Dulbecco′s Modified Eagle′s Medium (DMEM), supplemented with 10% Fetal Bovine Serum (FBS) and 1% penicillin, streptomycin and 1% glutamine for A549 and 1% amphotericin B for SW480. The A2780 and A2780cis were grown in RPMI 1640 supplemented with 10% Fetal Bovine Serum (FBS), 1% glutamine and 1% penicillin, streptomycin and amphotericin B. All cells were incubated in a humidified incubator at 37 °C under a 5% CO_2_ atmosphere and grown in 100 mm culture dishes to approximately 70–80% confluence. For subculturing, cells were detached with Trypsin-EDTA (Sigma). A549-gal were derived from A549 cultured in glucose-free medium supplemented with 10 mM galactose (Sigma) for at least 10 passes [54].

Cytotoxicity. MTT Assay and IC_50_ estimation. The cytotoxicity of **1[C,NH-Cl]**, **2[C,NMe-Cl]**, **3[N,NH-Cl]Cl**, **4[N,NMe-Cl]Cl**, **5[CNH_2_,NH-Cl]Cl**, **6[CNH_2_,S-Cl]Cl**, **7[CO,NH-Cl]** and **8[CO,S-Cl]** in A2780, A2780cis, SW480 and A549 carcinoma cells was evaluated by the MTT colorimetric assay. Cells were seeded in 96-well plates at 5000 cells per well and incubated at 37 °C, 5% CO_2_. After 24 h, the medium was removed and replaced with fresh medium with different concentrations of the compounds under study, and they were incubated for 48 h. Final DMSO concentration was kept below 0.7% in wells. For a negative control, one sequence of cells was left untreated and for vehicle control, cells treated with DMSO were included. For cell viability experiments preincubating with NAC (N-Acetyl-L-Cysteine), the cells were pre-treated with NAC, 10 mM for 2 h before the incubation with complex **1[C,NH-Cl]** for 24 h. Then, the medium was removed from the wells, and 100 µL MTT (5 mg/mL) was added. After an incubation time of 4 h, 100 μL of solubilizing detergent solution (SDS, 10% in HCL 0.01 M) was added and kept in the incubator overnight. Cell growth inhibition was detected by measuring the absorbance of each well at 595 nm using a BioTek Cytation5 microplate reader. Each test was repeated in quadruplicate in three independent experiments for each cell line. The percentage of survival cells was calculated using absorbance measurements compared with control and the IC_50_ estimation using Graph Pad Prism version 6.01.

Quantification of Ir cellular uptake by ICP-MS (Inductively coupled plasma-mass spectrometry). Metal accumulation studies were conducted on the SW480 cell line. In these experiments, 3 × 10^5^ cells were seeded in 12.5 cm^2^ tissue culture flasks in drug-free medium and incubated for 24h at 37 °C in a 5% CO_2_ humidified atmosphere. Then, cells were treated with 2 µM of the iridium complexes. After 24 h of exposure, cells were washed twice with PBS and harvested. The pellets were resuspended in PBS, and 10 µL were used to count cells. Then, each resuspended pellet was digested in 65% HNO_3_ for 24 h. Finally, solutions were analyzed in a 7700 ICP-MS (Agilent Technologies, Palo Alto, CA, USA). Data is reported as the mean the standard deviation (n = 3).

Native protein electrophoresis. The interaction between BSA and iridium complexes was studied using native polyacrylamide gel electrophoresis (PAGE). The BSA concentration was 3 µM and the C_D_/C_P_ ratio was 10. The samples of BSA/Ir-complexes were incubated overnight. The samples were mixed with a sample buffer (glycerol and bromophenol) and loaded into a 10% polyacrylamide gel for PAGE analysis (BioRad electrophoresis accessories). The gel was run at 80 V for 3 h, and then it was stained and destained before the gel was imaged using BioRad Gel Doc XR+ molecular imager. The staining solution contains 45% MeOH, 10% glacial acetic acid, 45% H_2_O and 0.1% brilliant blue, whereas the destaining solution consists of 20% MeOH, 10% glacial acetic acid and 70% H_2_O.

Specific assays with 1[C,NH] complex were performed according to manufacturer’s instructions using the following reagents from Molecular Probes (Thermo Fisher Scientific): Measurements of ROS: Dihydroethidium (DHE, D11347) and 2′, 7′-dichlorodihydrofluorescein diacetate (H_2_DCFDA, D399); detection of caspases 3/7: CellEvent Caspase-3/7 Green Detection Reagent (C10423); Cell viability: Live/Dead Cell Imaging Kit (R37601); Analysis of cytosolic calcium concentration: Fluo3/AM (F23915); Analysis of MMP: Tetramethylrhodamine methyl ester perchlorate (TMRM, T668). RT-qPCR’s assays were performed with the following primers/probes from Thermo Fisher Scientific: DCP1A (Hs00218198-m1), FOSB (Hs00171851-m1), EGR1(Hs0015928-m1), ATF3 (Hs00231069-m1), DDIT3 (Hs00358796-g1), PP1R15A (Hs00169585-m1), CHAC1(Hs00225520-m1), MIR9-1 (Hs04231512-s1), ATP1B2 (Hs00155922-m1), BBC3 (Hs00248075-m1), BCL2L11 (Hs00798019-s1), PMAIP1 (Hs00560402-m1), SKP2 (Hs01021864-m1) and HIST1H1B (Hs00271207-s1). RT-qPCR experiments were performed as described [55].

DNA damage was measured using EpiQuik 8-OHdG DNA Damage Quantification Direct Kit (colorimetric) (Fischer, Waltham, MA, USA, P600396); The mitochondrial cytochrome C release was evaluated using the “Millipore’s FlowCellect Cytochrome C Kit (Millipore, FCCH100110)”. Lactate dehydrogenase activity release was measured in the culture medium using Cytotoxicity Detection Kit (Roche Diagnostic, Barcelona, Spain, 11 644 793 001). Protein synthesis was determined using Click-iT Plus O-propargyl-puromycin (OPP) Protein Synthesis Assay Kit (Invitrogen™, Waltham, MA, USA). Measurement of GSH/GSSG was determined with GSH/GSSG-Glo Assay (Promega, V6611). Measurements of OCR and ECAR with Seahorse XFp (Agilent Technologies) were determined using the XFp Cell Mito Stress Test Kit (103010-100) and XFp Glycolysis Stress Test Kit (103017-100). ATP measurement was determined with Cell Titer-Glo Luminescent Cell Viability Assay (Promega, Madison, WI, USA, G7570). Determination of DNA damage using phosphorylated H_2_AX measurement were performed as previously described [56]. Cardiolipin oxidation using the fluorophore Nonyl Acridine Orange (NAO, Thermo Scientific, A1372) was performed as described [57]. Cell viability was determined by Hoechst inclusion and Propidium iodide (PI) exclusion by standard protocols [58]. TEM analysis were performed as described [59]. Cell cycle analysis by flow cytometry and microarrays analysis were performed as described [24]. Arrays data have been deposited in the ArrayExpress database at EMBL-EBI (www.ebi.ac.uk/arrayexpress) under accession number E-MTAB-860 (10 November 2022)

CHOP induction. HEK293 cells were seeded at 8 × 10^5^ cell/well in 24-well plates and transfected with CHOP::GFP plasmid (Addgene plasmid #21898; http://n2t.net/addgene:21898; RRID:Addgene_21898, 10 November 2022) using the transfection reagent DreamFectTM Gold (OZ Bioscience, Cat. N. º LM80250). After 24 h incubation, the treatments were added: **1[C,NH]** 9.7 µM; tunicamycin 0.5 µM as positive control. Cells were stained with propidium iodide (1 µg/mL). Time-lapse images were acquired every 30 min for 24 h.

Measurements. Absorbance and luminescence emission intensity were recorded by CLARIOstar^®^ BMG Labtech, software Mars. Flow cytometry analyses were determined by a Guava EasyCite, Millipore. Fluorescence images were obtained by IX51 SBF-3 Olympus fluorescence microscopy, Software Cell Sense. Time Lapse imaging were performed using Wide Field Okolab DMI600B, DFC360FX camera, controlling by CTR7000HS, software LAS AF (Leica).

In vivo studies.

Female mice weighing approximately 20–25 g, 8–12 weeks were obtained from Janvier Laboratories (Le Genest-Saint-Isle, France). All mice received a standard laboratory diet and water ad libitum. The use of immunosuppressed animals is always a limitation since they are not a true reflection of an animal with a functional immune system and, therefore, can affect both the toxicological response and the biodistribution of a compound, but they are essential to be able to implant human tumor cells, otherwise the tumor would be rejected by the immune system. Therefore, in order to analyze the biodistribution, toxicology and efficacy of a compound against a tumor, it is necessary to use different animals. Animal studies were approved by the Bioethics Committee of the Universidade de Santiago de Compostela and were done in compliance with the principles of laboratory animal care according to Spanish national law (RD 53/2013).

Biodistribution studies: BALB/c mice were injected via the tail vein with 100 µL of **1[C,NH]** dissolved in PBS. Then, 4 h after the injection, the animals (n = 3) were sacrificed, and the following organs were removed: kidney, spleen, liver, heart, lung, pancreas, intestine, ovary and stomach. They were kept at −80 °C until their analysis at the Cactus-Lugo Instrumental Analysis Unit. Solutions were diluted to a final volume of 10 mL MilliQ water and measured on an Agilent 7700x-ICP-MS.

Orthotopic lung cancer model. Athymic nude mice were used. The model was developed following the protocol previously described [60,61]. On day 20, treatment with the metal complex was started, and 4 doses were given every other day. Tumor growth was measured in vivo by luminescence (IVIS^®^ Spectrum) on the days that the complexes were not administered, ending the experiment on day 37 with the sacrifice of the animal and the removal of organs. Organ indicators of metastases, lungs, and mediastinal lymph nodes were processed as previously described [62]. Histological analyzes were performed as described [24].

Statistical analysis. The program used for statistical analysis was Graph Pad Prism version 6.01. The *p*-values were calculated using the *t*-test (* *p* < 0.05, ** *p* < 0.01, *** *p* < 0.001).

## 3. Results and Discussion

Synthesis and characterization. Eight Ir(III) half-sandwich complexes were synthesized and fully characterized. Methylated ligands [48] and the Ir(III) starting material [IrCl(µ-Cl)(η^5^-Cp*)]_2_ [63] were obtained from adapted protocols in the literature. The cationic Ir(III) complexes [IrCl(η^5^-Cp*)(κ^2^-N,N-L3)]Cl (**3[N,NH-Cl]Cl**), [IrCl(η^5^-Cp*)(κ^2^-N,N-L4)]Cl (**4[N,NMe-Cl]Cl**), [IrCl(η^5^-Cp*)(κ^2^-N,N-L5)]Cl (**5[CNH_2_,NH]Cl**) and [IrCl(η^5^-Cp*)(κ^2^-N,N-L6)]Cl (**6[CNH_2_,S-Cl]Cl**) were prepared by reacting [IrCl(µ-Cl)(η^5^-Cp*)]_2_ with the respective ligand in CH_2_Cl_2_ while the neutral derivatives [IrCl(η^5^-Cp*)(κ^2^-C,N-L1)] (**1[C,NH-Cl]**), [IrCl(η^5^-Cp*)(κ^2^-C,N-L2)] (**2[C,NMe-Cl]**), [IrCl(η^5^-Cp*)(κ^2^-N,O-L7)] (**7[CO,NH-Cl]**) and [IrCl(η^5^-Cp*)(κ^2^-N,O-L8)] (**8[CO,S-Cl]**) were synthesized by reacting [IrCl(µ-Cl)(η^5^-Cp*)]_2_ with the appropriate pro-ligand in the presence of NaOAc (**1[C,NH-Cl]** and **2[C,NMe-Cl]**) or Et_3_N (**7[CO,NH-Cl]** and **8[CO,S-Cl]**). A scheme for the synthesis of the complexes is shown in Figure 2, and the details are given in the Appendix A. Some of the complexes were previously reported; a derivative of complex **3[N,NH-Cl]^+^** bearing PF_6_^−^ as the counterion was reported by Kollipara [64], the aqua complex **3[N,NH-H_2_O]^2+^** was published as the sulphate salt by Himeda [65], the crystal structure of complex **3[N,NH-Cl]** with PF_6_^−^ as counterion was reported by Chellan [66] and complex **6[CNH_2_,S-Cl]** was published by D. S. Pandey [67]. All complexes were isolated as the corresponding racemates (RIr or SIr) in moderate or good yields (from 38% to 98%). The complexes were fully characterized by NMR spectroscopy (1D and 2D), IR spectroscopy, positive fast atom bombardment (FAB+) mass spectrometry (see Appendix A), molar conductivity and elemental analysis. Purity was determined by analytical HPLC, confirming ≥95% (Appendix A). For additional information about molar conductivity experiments, see Appendix A and Appendix A.

Single crystals suitable for X-ray diffraction analysis were obtained for **1[C,NH-Cl]**, **3[N,NH-Cl]Cl·2H_2_O**, **4[N,NMe-Cl]·H_2_O**, **5[CNH_2_,NH-Cl]·H_2_O** and **8[CO,S-Cl]** by the slow evaporation of solvents or a combination of solvents: methanol/acetone, methanol/water and methanol/dichloromethane [15,17]. The structure of **5[CNH_2_,NH]** was reported by D. S. Pandley et al [67]. A summary of selected bond lengths and angles as well as selected geometric parameters is given in Appendix A. The crystal data collection and refinement parameters are gathered in Appendix A.

The ORTEP diagrams for all the complexes are represented in Figure 3, and the unit cells show the expected enantiomers (R_Ir_ and S_Ir_) with the pseudo-octahedral three-legged piano-stool geometry and the iridium π-bonded to a η^5^-Cp* (pentamethylcyclopentadienyl ring). The Ir-centroid (1.766–1.825 Å) and Ir-Cl (2.385–2.422 Å) distances are standard. The Ir-N(benzimidazole, bim or benzothiazole, btz) distances fall in a narrow interval (2.037–2.098 Å), whereas the Ir-C/N/O(phenyl, pyridyl, aminophenyl or hydroxyphenyl) distances are longer (2.097–2.202 Å).

The bite angles (75–80°) are determined by the features of the corresponding free bidentate ligands and tend to be bigger than those obtained for their Ru(II) congeners (76–77°) [68,69]. Moreover, complexes of series A and B have 5-membered chelate rings, and derivatives of series C and D have 6-membered chelate rings. The metallacycle size has an impact on the geometry of the corresponding rings. Thus, complexes **1[C,NH-Cl]**, **3[N,NH-Cl]·2H_2_O**, **4[N,NMe-Cl]·H_2_O** possess planar geometries, whereas complexes **5[CNH_2_,NH-Cl]·H_2_O** and **8[CO,S-Cl]** exhibit an envelope conformation for the respective metallacycles, as it can be deduced from the torsion angles (θ, Appendix A).

The 3D structure of the complexes is supported by strong intermolecular hydrogen bonding interactions (between co-crystallized water molecules and the NH group of the benzimidazole moiety or the NH moiety and the chloride) as well as weak (C-H···π and π-π stacking) interactions. Among these interactions, it is remarkable that the π-π stacking was between the Cp* and the benzene ring of benzothiazole moiety in complex **8[CO,S-Cl]**, which is reinforced by the C-H···π interactions of the methyl groups of Cp* and the same benzene ring (Appendix A and Appendix A). Although these interactions involving Cp* are unusual and weak, some examples in complexes with other metal ions have been reported in the literature [70,71,72,73].

Solubility, aquation equilibria and acid-base behavior. Water solubility studies and analysis of the aquation/solvolysis equilibria were performed to decipher the behavior of these complexes in biological media. Solubility studies of this family of complexes revealed a strong dependence on the bidentate ligand as well as on the charge of the complex. Thus, the Ir-N^N and Ir-N^NH_2_ complexes (**3[N,NH-Cl]^+^**, **4[N,NMe-Cl]^+^**, **5[CNH_2_,NH-Cl]^+^** and **6[CNH_2_,S-Cl]^+^**) are monocationic and water-soluble, whereas Ir-N^C, (**1[C,NH-Cl]** and **2[C,NMe-Cl]**) and Ir-N^O (**7[CO,NH-Cl**] and **8[CO,S-Cl]**) are neutral and soluble in DMSO, but not in water. All complexes fulfil Lambert-Beer law (Appendix A), and no-aggregates formation was observed. Regarding the stability of the complexes in aqueous solutions, the water soluble complexes undergo straightforward aquation, leading to the replacement of chloride by a water molecule, to give **3[N,NH-H_2_O]^2+^**, **4[N,NMe-H_2_O]^2+^**, **5[CNH_2_,NH-H_2_O]^2+^** and **6[CNH_2_,S-H_2_O]^2+^**. The kinetic spectrograms and the data pairs absorbance-time at a fixed wavelength are displayed in Appendix A. The corresponding curves were fitted to monoexponential functions to obtain the rate constants kaq (see Appendix A). Aquation may facilitate DNA binding reactions [74,75].

Regarding the neutral complexes **1[C,NH-Cl]**, **2[C,NMe-Cl]**, **7[CO,NH-Cl]** and **8[CO,S-Cl]**, they underwent substitution of the chloride when dissolved in DMSO to give **1[C,NH-DMSO]^+^**, **2[C,NMe-DMSO]^+^**, **7[CO,NH-DMSO]^+^** and **8[CO,S-DMSO]^+^**) as confirmed by UV-vis (Appendix A) and NMR (Appendix A). Furthermore, the initial suspension became colorless as the reaction evolved (Appendix A). Strong affinity of iridium(III) complexes to bind dimethyl sulfoxide is well known [8]. From the fitting of a monoexponential function to Abs-time data pairs (insert Appendix A), we obtained the k_DMSO_ rate constants (see Appendix A). The kinetic constants k_aq_ and k_DMSO_ of formation of **6[CNH_2_,S- H_2_O]^2+^** and **8[CO,S-DMSO]^+^** are (2.12 ± 0.01) × 10^−4^ s^−1^ and (3.11 ± 0.25) × 10^−4^ s^−1^, respectively, between 10 and 100 times slower than k_aq_ and k_DMSO_ of the other complexes. This behavior indicates that the Cl-Ir bond is stabilized by the presence of benzothiazole as the ancillary ligand compared to benzimidazole.

The acid–base behavior is relevant since the protonation state of the synthesized compounds modulates not only the establishment of supramolecular interactions with biomolecules, but also their subcellular localization. The acid dissociation constants were calculated from the absorbance spectra at different pH. UV-vis spectrograms and absorbance at a fixed λ-pH data pairs are shown in Appendix A. The Henderson–Hasselbalch [76] equation was applied for the complexes that exhibited one acid-base equilibria, **1[C,NH-DMSO]^+^** (Appendix A), **4[N,NMe-H_2_O]^2+^** (Appendix A), **6[CNH_2_,S-H_2_O]^2+^**(Appendix A) and **7[CO,NH-DMSO]^+^** (Appendix A). The Abs vs. pH plot for **3[N,NH-H_2_O]^2+^**(Appendix A) showed a minimum at pH ≈ 7 and for **5[CNH_2_,NH-H_2_O]^2+^** (Appendix A) a maximum at pH ≈ 9, denoting, in both cases, two consecutive acid-base equilibria that were analyzed according to the Ang equation [77]. Finally and as expected, **2[C,NMe-DMSO]^+^** (Appendix A) did not exhibit any acid–base equilibrium since there are not susceptible atoms prone to undergo this kind of equilibrium. The calculated pK_a1_ and pK_a2_ values are gathered in Appendix A. At pH >10, the benzothiazole complex **8[CO,S-DMSO]^+^** (Appendix A) underwent fast substitution of DMSO by OH^−^ to give **8[CO,S-OH]** complex as it has been confirmed by stopped flow measurements (Appendix A). pK_a1_ values of the soluble compounds **3[N,NH-H_2_O]^2+^, 4[N,NMe-H_2_O]^2+^, 5[CNH_2_,NH-H_2_O]^2+^** and **6[CNH_2_,S-H_2_O]^2+^** correspond to the dissociation of an hydrogen atom from water to form the monohydroxo species [17,78]. Two ionization equilibrium constants of 2-phenylbenzimidazol ligand have been calculated, which correspond to dissociation of the pyridine- (=NH^+^) and pyrrole-type (–NH) hydrogen atoms (Scheme S1). The first pK = 5.2 ± 0.1 is related to the deprotonation of the pyridine type N atom, and the pK = 11.8 ± 0.1 is assigned to that of the pyrrole type N atom. Therefore, pKa_2_ corresponds to NH dissociation in **1[C,NH-DMSO], 3[N,NH-H_2_O]^2+^ 5[CNH_2_,NH-H_2_O]^2+^** and **7[CO,NH- Cl]** [79,80]. In view of these results, we can conclude that at pH 7 the complexes are mostly found as in Appendix A with the exception of 3, which will be found mostly as **3[N,NH-OH]^+^**. As we can observed, **X** and **Z** play an essential role on the acid-base behavior of the different iridium complexes under study. From now on, the complexes will be denoted as **n**[**Z**,**X**] for simplification. 

**Cytotoxic activity and cell accumulation.** The cytotoxicity of the complexes was evaluated in representative human cancer cells: A2780 and A2780cis (epithelial ovarian carcinoma and acquired resistance to cisplatin), SW480 (colon adenocarcinoma) and A549 (epithelial lung adenocarcinoma) cells. The cytotoxicity of cisplatin was also evaluated as a reference. The results, gathered in Figure 4A and Appendix A, show some interesting structure–activity relationships.

We can observe that the atom in the **Z** position plays an essential role in the cytotoxic activity of these derivatives. Complexes with atoms negatively charged directly bonded to iridium, **Z** = C and CO (**1[C,NH]**, **2[C,NMe]** and **8[CO,S]**) are more cytotoxic than those with neutral atoms attached to the metal center, **Z** = N and CNH_2_ (**3[N,NH]**, **4[N,NMe]** and **6[CNH_2_,S]**). The only exception is **7[CO,NH],** which is less cytotoxic than **5[CNH_2_,NH]** in SW480 cells. Regarding **X**, we have observed that, in general, the NH group increases the activity when compared to the N-Me group or the S atom (**1[C,NH]** vs. **2[C,NMe]**, **3[N,NH]** vs. **4[N,NMe]**, **5[CNH_2_,NH]** vs. **6[CNH_2_,S]**). The only exception is **7[CO,NH],** which is less cytotoxic than **8[CO,S]**. In conclusion, different substituents on the **Z** and **X** positions modulate the cytotoxicity of the arylbenzazole-based iridium(III) complexes. Overall, the less cytotoxic complex in all the cell lines is **4[N,NMe]**, whereas the most cytotoxic derivative is **1[C,NH]** except in A2780 where **8[CO,S]** is more active. In addition, **1[C,NH]** is more cytotoxic than cisplatin in the four tested cancer cell lines, which confers a special biological relevance to this compound.

Finally, the Resistance Factor (RF) calculated as IC_50_,_A2780CIS_/IC_50,A2780_ (Figure 4B) reveals that complexes **7[CO,NH]** and **8[CO,S]** are not able to circumvent cisplatin acquired resistance on A2780cis since RF > 2 [81]. By contrast, the best performers in all the studied cell lines, complexes **1[C,NH]** and **2[C,NMe],** exhibited no cross-resistance.

Cellular uptake of the complexes was measured through the determination of the iridium concentration inside SW480 cells by ICP-MS after 24 h of incubation with 2 μM of the complexes (Figure 4D). All of them, except **6[CNH_2_,S]**, show better accumulation than cisplatin, according to the following trend: **8[CO,S]** > **1[C,NH]** > **2[C,NMe]** > **5[CNH_2_,NH]** > **7[CO,NH]** > **3[N,NH]** > **4[N,NMe]** > **cisPt** > **6[CNH_2_,S]**. Nonetheless, seroalbumin, the most representative example among proteins in blood plasma and in cell culture media, could sequestrate the complexes and thus lead to lower concentrations inside cells [82]. Native Polyacrylamide Gel Electrophoresis (PAGE) can be used to study the interactions of a metal complex with bovine seroalbumin, BSA. Figure 4C shows that **1[C,NH]**, **2[C,NMe]**, **5[CNH_2_,NH]** and **6[CNH_2_,S]** do not interact with the protein, so BSA does not hinder their uptake by the cells. The interactions of **3[N,NH]**, **4[N,NMe]** and **7[CO,NH]** with BSA suggest that these complexes can be sequestrated by the protein, hindering their cellular accumulation, which might be a possible explanation for their low cytotoxicity. Surprisingly, **8[CO,S]** interacts with BSA, but it still exhibits the highest accumulation in SW480 and a high cytotoxic activity. Thus, BSA binding do not explain/justify the differences observed in the cytotoxicity of the complexes.

DNA binding. According to literature data, the mechanism of action of half-sandwich Ir(III) anticancer complexes may involve both interaction with DNA and perturbation of the redox status of cells [12,19,83,84]. Since DNA-binding cytotoxic drugs are the first choice in the treatment of many cancers [85,86,87,88,89], NMR [69], UV-Vis and Circular Dichroism (CD) spectroscopies, viscosity [90,91,92] and Differential Scanning Calorimetry (DSC) experiments were performed to properly check DNA as a possible target and to clarify the differences in the cytotoxicity (see in DNA interaction section in the Appendix A). UV-Vis and NMR spectroscopies (Appendix A) allowed us to conclude that complexes **3[N,NH]**, **4[N,NMe]**, **5[CNH_2_,NH]**, **6[CNH_2_,S]**, **7[CO,NH]** and **8[CO,S]** are able to covalently bind DNA. Furthermore, the mass spectra of **3[N,NH]** with 9MeG and GMP (as simplified models for DNA interaction) revealed coordination to both of them (see Appendix A). The corresponding kinetic rate constants, kc, are shown in Appendix A. Nevertheless, DNA covalent binding can be ruled out for **1[C,NH]** and **2[C,NMe]**. To corroborate if these two complexes are able to bind DNA in a non-covalent way, absorbance titrations were carried out. No variation in the recorded spectra upon the addition of DNA was observed (Appendix A). Relative viscosities (Appendix A) and melting temperature, Tm, (determined by differential scanning calorimetry) (Appendix A) were also measured for each C_D_/C_P_ ratio (D and P being the concentration of complex and DNA, respectively) after overnight incubation in order to ensure the formation of the covalent Ir-DNA bond. No variations were observed for **1[C,NH]** and **2[C,NMe]**, confirming again no DNA interaction. Finally, circular dichroism spectrograms were recorded at different C_D_/C_P_ ratios after overnight incubation to allow the covalent Ir-DNA bond formation. It is worth mentioning that our complexes have been isolated as the respective racemic mixtures, which are optically inactive. The evolution of CD bands gave final spectra showing important differences (Figure 5 and Appendix A). The new positive and negative induced circular dichroism (ICD) bands account for the formation of Ir-DNA adducts [93]. Once more, it can be concluded that **1[C,NH]** does not bind to DNA since there are no variations in CD spectra. In addition, for **2[C,NMe]**/DNA system, the recorded CD spectra show no significant variations, which is compatible with an external binding [94]. Nonetheless, the variation in CD spectrograms at different C_D_/C_P_ ratios of **3[N,NH]**, **4[N,NMe]**, **5[CNH_2_,NH]**, **6[CNH_2_,S]**, **7[CO,NH]** and **8[CO,S]** are prominent. Thus, with the exception of **8[CO,S]**, the interaction of the complexes with DNA does not account for differences in cytotoxicity. In addition, the most cytotoxic compound **1[C,NH]** does not interact with DNA (Figure 5A). The less cytotoxic derivatives, **3[N,NH]** and **4[N,NMe]**, generate strong changes in the DNA structure as a consequence of the interaction (Figure 5B and Appendix A). Nevertheless, their toxicity is low due to their limited cellular accumulation (Figure 4D). In general, the increase of molar ellipticity [θ] in the ICD region with C_D_/C_P_ corresponds to an intercalation process [65,95], and a decrease corresponds to groove binding [96]. However, the alternation in the sign of ICD bands obtained in all experiments, rules out any interpretation about the specific binding mode and reveals the intricacy of the systems. The changes in the relative viscosity (Appendix A) and melting temperature (Appendix A) as a function of the C_D_/C_P_ ratio suggest that a bifunctional covalent-groove binding product is the most likely DNA binding mode [97].

The gel electrophoresis assay (Appendix A) confirms previous results, that is, complexes **1[C,NH]** and **2[C,NMe]** are unable to interact with DNA since no changes in the migration pattern of the plasmid occur. By contrast, the rest of the complexes alter this migration pattern confirming DNA binding.

ROS generation. Several cyclopentadienyl Ir(III) complexes able to alter ROS levels and lead cancer cells to death have been previously reported [8,19,84,98,99,100]. Hence, the ability of complexes to induce ROS was investigated (Appendix A). We used two fluorescent probes: dihydroethidium (DHE) to screen for superoxide anion (O_2_^∙−^) levels and 2′, 7′-dichlorodihydrofluorescein diacetate (H_2_DCFDA) to detect hydrogen peroxide (H_2_O_2_) among other oxidative species, such as nitric oxide and peroxynitrite [101]. After 3 h, DHE fluorescence increased for complexes **1[C,NH], 2[C,NMe]** and **8[CO,S]**. H_2_DCFDA fluorescence also increased, supporting the view that those complexes cause the increase of O_2_^∙−^ levels (See Appendix A).

Mitochondrial dysfunction induced by **1[C,NH]** complex. Therefore, **1[C,NH]** increases ROS levels (Figure 6A,B), does not bind to DNA, and is cytotoxic in all studied cell lines. This peculiarity, together with its resistance factor, led us to select this complex to study in more detail its possible mechanism of action. We were aware of the limitations of using fluorescent probes to estimate ROS levels [102], so we studied the effects of **1[C,NH]** on well-known ROS targets [103]. DNA damage was detected using a phosphorylated histone marker (γH2AX) by flow cytometry [24]. A dose-dependent increase in DNA double-strand breaks was seen when cells were incubated with **1[C,NH]** (Figure 6C). Moreover, 8-hydroxy-2′-deoxyguanosine (8-OHdG) levels, an oxidized derivative of deoxyguanosine [104] was increased after 4 h (Appendix A). The determination of the reduced glutathione (GSH)/oxidized glutathione (GSSG) ratio is a useful indicator of oxidative stress in cells and tissues. **1[C,NH]** decreased the GSH/GSSG ratio as expected (Appendix A). To explore if the diminished GSH/GSSG ratio had biological consequences, we tested whether the cytotoxicity induced by **1[C,NH]** was rescued with N-acetyl-cysteine (NAC) coadministration. The mechanism of NAC action is debatable. However, it is currently accepted that NAC chemical interaction with hydroperoxides is extremely unlikely, as NAC acts as precursor of GSH in conditions of extreme GSH depletion [105]. In agreement with a GSH mediated NAC action, we found a partial rescue from cytotoxic effects of **1[C, NH]** in the presence of NAC (Appendix A).

Proteins and lipids that integrate the mitochondrial membrane may be oxidized by H_2_O_2_. Cardiolipin(CL)-bound cytochrome c (Cyt c) acts as a peroxidase capable of catalyzing H_2_O_2_-dependent CL peroxidation, which is an essential step in the release of Cyt c during apoptosis.[62] **1[C,NH]** induced CL oxidation (Figure 6D) and increased Cyt c release (Figure 6E). Moreover, the administration of **1[C,NH]** induced caspase 3/7 activation (Figure 6F) (see Appendix A). All this data indicated that **1[C,NH]** prompts a form of programmed cell death. Morphological changes presented by the cells were also compatible with the process of apoptosis [106]: retraction of the cytoplasm, nuclear fragmentation and formation of apoptotic bodies. Furthermore, some cells showed secondary necrosis, with a balloon-like morphology. All in all, these events are the natural outcome of the complete apoptotic program (Appendix A) [107]. Swollen ER and mitochondria, early signals of preapoptotic cells [108] and the presence of large invaginations of the plasma membrane were present 2 h after the treatment. After 12 h numerous autophagosomes, as a signal of mitochondrial affectation [109], were detected (Appendix A). Hoechst/Propidium Iodide (PI) staining was used to exclude necrosis [110] as the principal cause of death. PI positive cells were absent during the first 6 h of treatment and were only present 24 h later, associated with secondary necrosis (Appendix A). Furthermore, release of lactate dehydrogenase (LDH) to the medium, owing to the altered permeability of the plasmatic membrane, only began after 8 h of incubation (Appendix A).

Studies of the effect of **1[C,NH]** on gene expression were performed and assessed 6 h after treatment, in a middle phase, when most cells were still alive (see Appendix A) but already with characteristic morphological features of apoptosis (Movie S1). Gene expression profiles of cells was determined by Affymetrix Human Gene 2.1 ST Array Plate in three independent experiments. In order to confirm the microarray data, we selected some genes involved in several key pathways: stress response, apoptosis and cell cycle to validate the changes in the expression by RT-qPCR. Results indicated a good correlation between microarray analysis and RT-qPCR. (Appendix A). We next used Gene Ontology (GO) enrichment analysis to compare the biological processes that were significantly affected. Using GO term finder combined with ReviGO, we classified the function of all gene hits whose expression was significantly increased by **1[C,NH]** into the broad categories of (i) “Negative regulation of cellular macromolecule biosynthetic process,” (ii) “negative regulation of cellular metabolic process,” (iii) “negative regulation of macromolecule metabolic process” and/or (iv) “negative regulation of cellular process”. Note that not all genes are annotated with these databases. However, the negative regulation of metabolic process and negative regulation of macromolecule metabolic process are highly represented, and ~38% (84) of all genes could be classified in this way. On the other hand, **1[C,NH]** significantly decreased the expression of histone genes, consistent with the fact that treatment with **1[C,NH]** induces a stress response that leads to the arrest of the cell cycle and the entry into apoptosis. The effect of **1[C,NH]** on cell cycle progression was further assessed by flow cytometry. Then, 24 h after treatment, the percentage of cells in G0/G1 phase increased and cells in G2/M phase decreased (Appendix A), thus justifying the down regulation of G2/M phase genes. In summary, mRNA expression analysis is in good agreement with the elevation of ROS, oxidative stress response, cell cycle arrest and induction of apoptosis.

Since all these experiments point out at an increment of ROS levels for **1[C,NH]**, we wondered about the initial source of ROS. We explored first whether **1[C,NH]** directly affected mitochondrial respiration. Oxygen consumption rate (OCR) provides a measurement of mitochondrial respiration. After treatment of A549 cells with **1[C,NH],** basal OCR was not affected. Interestingly, when oligomycin (Oligo), an ATP synthase inhibitor, was added to **1[C,NH]** treated cells, OCR was not affected, indicating that the main component of OCR in the treated cells was not linked to ATP production (Figure 7A). Therefore, we tentatively concluded that administration of **1[C,NH]** increased proton leaks (Appendix A). When the protonophore FCCP (carbonyl cyanide p-trifluoromethoxyphenyl hydrazone) was added, OCR remained unchanged again instead of raising the OCR to the maximum values. Again, this suggests that complex **1[C,NH]** blocks the ability of the electron transport chain (ETC) enzymatic system to reduce O_2_ and generate a proton gradient through the mitochondrial membrane. Since the mitochondrial membrane potential (MMP) seemed to be affected by our compound, MMP was evaluated using the fluorescent potentiometric dye, tetramethylrhodamine methyl ester (TMRM). As seen in Figure 7E, **1[C,NH]** induced a quick and deep dissipation of MMP confirming again that administration of **1[C,NH]** increases proton leaks. To test the effect in the non-mitochondrial respiration, mitochondrial respiration was blocked with a mixture of rotenone (R, a complex I inhibitor), and antimycin A (A, a complex III inhibitor). OCR in **1[C,NH]** treated cells was reduced to the levels of control cells, demonstrating that non-mitochondrial OCR was not affected by **1[C,NH]** administration (Figure 7A). Thus, **1[C,NH]** selectively disrupts mitochondrial respiration. Therefore, **1[C,NH]** was selectively acting on mitochondrial respiration.

The mitochondrial capacity to produce ATP dropped markedly in cells treated with **1[C,NH]** (Appendix A). However, a small decrease of around 35% on ATP levels was found after 8 h incubation with **1[C,NH]** (Appendix A). Accordingly, cells needed to harness the glycolytic pathway in order to maintain ATP levels, as seen by the increase in the extracellular acidification rate (ECAR), that is largely the result of glycolysis (Figure 7B). Moreover, the modulation of mitochondrial activity with Oligo, FCCP and R&A had no effect on ECAR, supporting that **1[C,NH]** disrupted mitochondrial ATP synthesis. Oxidation of galactose to pyruvate yields no net ATP, which prompts cells to oxidize pyruvate via OXPHOS (oxidative phosphorylation) to survive. Thus, we used A549 galactose-grown cells (A549-gal) [54] to further explore the effects of **1[C,NH]**. A small increase in OCR was detected after addition of **1[C,NH],** as expected for an uncoupled agent. Nevertheless and taking into account previous data, no variation in the OCR values was found in **1[C,NH]** treated A549-gal cells after Oligo addition since mitochondria ATP synthesis was disrupted due to **1[C,NH]**treatment (Figure 7C). Furthermore, confirmation that **1[C,NH]** did not affect glycolysis, was obtained when glucose was offered to A549-gal cells a seen by the huge ECAR increase in **1[C,NH]** treated cells (Figure 7D). Moreover, when treated cells were offered 2-DG (2-Deoxy-D-glucose), that blocks glucose metabolism, ECAR immediately dropped indicating that glycolysis was the major contributor to ECAR in treated cells (Figure 7D) (Appendix A). In summary, cells endure the dysfunction of the mitochondria increasing their glycolytic capacity to maintain the ATP levels. **1[C,NH]** also presented similar effects in a non-tumor cell line, MCF10A.

Mitochondrial dysfunction triggers the integrated stress response (ISR). An increase in cytosolic Ca^2+^ was also observed after 4 h of incubation (Appendix A). To gather more evidence, we assessed the effect of **1[C,NH]** on protein synthesis. ISR is known to trigger eIF2α phosphorylation that inhibits protein synthesis. After 6 h, **1[C,NH]** inhibited around 60% of the total protein synthesis (Appendix A). Furthermore, we also observed an increased expression of the ER stress-responsive transcription factor C/EBP homologous protein (CHOP) (Appendix A) thus confirming that **1[C,NH]** was triggering the ISR as result of **1[C,NH]** causing mitochondrial dysfunction.

**1[C,NH]** selective action on mitochondrial bioenergetic, inducing oxidative stress, led us to explore it as an antitumor drug. It is generally accepted that apoptosis is initiated when ROS generation overflows the cellular antioxidant pathways. We reasoned that proliferating cells should be more sensitive to **1[C,NH]** action rather than terminally differentiated, non-proliferating normal cells because ROS production is different in both conditions. Cell division raises the energy demand and as a by-product, mitochondrial ROS production is increased. Hence, **1[C,NH]** proapoptotic effect should be greater in proliferating cells with an already elevated ROS production. We have addressed this point by assessing complex **1[C,NH]** action in the same cell line under proliferating (increased ROS production) vs. non-proliferating conditions. Significantly, we found that the IC_50_ of the complex **1[C,NH]** was 9.5 µM in the proliferating condition vs. 17 µM in the non-proliferating condition (see Figure 7F). Thus, confirming our prediction and opening the therapeutic window for the complex. In summary, complex **1[C,NH]** acts enhancing the oxidative stress of the tumor cell to induce apoptosis. Remarkably, normal vs. mutated oncogenes are non-longer relevant for complex **1[C,NH]** action, unless driver oncogenes modify the metabolism of the tumor cells, as it frequently happens, inducing cell proliferation and, as a by-product of the increased energetic demand, oxidative stress.

**1[C,NH]** reduced the lung tumor burden. Prior the performance of in vivo studies, the drug uptake was determined. The amount of **1[C,NH]** entering the cell was estimated after administration of 16 μM of **1[C,NH]** to 400,000 cells. The intracellular iridium content was assessed after 4 h by ICP-MS, and the percentage of drug taken up by cells was 1.7%. Then, acute toxicity and bio distribution of **1[C,NH]** was studied in healthy mice (BALB/c). **1[C,NH]** was dissolved in PBS The maximum tolerated dose was (1.3 mg/kg) and the bio distribution showed that most irrigated tissues were the ones with the highest accumulation of **1[C,NH],** such as liver, kidneys and lungs (Figure 8A). Antitumor efficacy was studied using an orthotopic model of lung cancer in nude mice [24] (Figure 8B). A549-luc cells (A549 cells marked with luciferase) were used to induce the tumor and 20 days after the injection of cells, we proceeded to the administration of 4 doses (0.9 mg/kg) of **1[C,NH]** on alternate days. Treated mice did not lose weight. During the necropsy, no pathological evidence of relevance was detected in non-tumor tissues in treated animals. However, we cannot exclude the appearance of toxic effects after repeated administrations over longer periods of time. Control animals showed a steady tumor growth up to the end of the experiment, 37 days after injection. **1[C,NH]** had no effect on the reduction of tumor burden, as seen by daily measurements of the bioluminescence in tumor cells. However, this estimation did not discriminate between fluorescence coming from the primary tumor and from the affected lymphatic nodes. To differentiate the effect of **1[C,NH]** on the primary tumor and on the affected nodes, at day 37, the animals were sacrificed, and luciferase activity, as a marker on the presence of tumor cells, was measured. Interestingly, **1[C,NH]** significantly reduced luciferase activity in the primary tumor but not in the neighbor invaded mediastinal nodes (Figure 8C). Moreover, lung mice were stained with a monoclonal antibody against a human cytokeratin (only A549 tumor cells of human origin were stained) (Figure 8D). Notably, **1[C,NH]** had a strong effect on reducing the size of tumor foci, confirming that **1[C,NH]** had a strong antitumor action on the primary tumor. The lack of effect on mediastinal nodes could be related with a low penetration of **1[C,NH]** into the lymphatic nodes. In addition, other factors influence the efficacy of a drug on the primary tumor and not on the metastasis, cell-cell interactions, matrix in the nodes, etc. However, **1[C,NH]** differential effects on the primary tumor and in affected mediastinal nodes could be the result of mitochondrial plasticity in tumor progression [111]. Cancer cells continuously rewire their metabolism to fulfil their need for rapid growth and survival while subject to changes in environmental cues. Cancer cells that depend mainly on glucose for ATP synthesis, such as hypoxic cells, depend less on OXPHOS and are thus more resistant to **1[C,NH]**. We believe, therefore, that the sensitivity or resistance to **1[C,NH]** is given by the activity of the mitochondria that will vary depending on the tumor and its location.

## 4. Conclusions

A family of eight arylbenzazole-based iridium(III) complexes have been rationally designed, prepared and characterized, in order to set a SAR study. Substitutions on the atoms of the ancillary ligand led to changes in the global charge of the complex, modulating their solubility and considerably affecting their biological properties. 

The SAR study revealed that complexes with N^N neutral chelating ligands are water soluble and form aqua complexes. However, the neutral complexes bearing anionic C^N and O^N ligands are soluble in DMSO and not in water, leading to the replacement of the chloride leaving group by DMSO. Regarding the cytotoxic activity, all the complexes showed activity in the complexes bearing the anionic ligands C^N and O^N are more cytotoxic than those with N^N ligands. In addition, the NH moiety of the imidazole seems to play an important role in the cytotoxicity when compared with the N-methylated imidazole or thiazole. All the complexes, except **1[C,NH]** and **2[C,NMe]**, bind DNA. In general, neutral complexes, especially those bearing the phenylbenzimidazole ligand, internalize better, do not bind easily with biomolecules and are cytotoxic.

Among all the complexes studied we selected the one with the highest cytotoxicity and selectivity, **1[C,NH]** for further characterization. It is generally accepted that ETC dysfunction causes increased O_2_^∙−^ levels. In the mitochondria, superoxide dismutase catalyzes the conversion of O_2_^∙−^ into H_2_O_2_ and O_2_. H_2_O_2_ molecules are eliminated by thiol peroxidases, such as glutathione peroxidases and peroxiredoxin 3. When H_2_O_2_ generation overflows these antioxidant pathways initiates apoptosis. Accordingly, we found **1[C,NH]** toxic effect could be abrogated by the thiol-reducing agent, NAC. Consequently, we found that **1[C,NH]** induced CL oxidation, and as a result of it, the mobilization of Cyt c from CL, allowing for Cyt c release and consequent activation of caspases 3/7, ultimately leads to apoptotic cell dismantling. Whether **1[C,NH]** increases electron leaks remains to be established.

Moreover, mitochondria remain the most critical location where ATP is produced. Thus, a malfunction of the ETC can provoke cell dead due to ATP depletion. Herein we show that **1[C,NH]** increased proton leaks; mitochondrial proton leak is the principal mechanism that incompletely couples substrate oxygen to ATP generation, thus explaining the ATP decrease found after treatment with **1[C,NH]**. There are contradictory reports that link proton leak to ROS generation. However, the view that uncoupling decreases mitochondrial ROS production prevails. Therefore, uncoupling may play a protective role by mitigating ROS production in cells. Hence, we cannot exclude that proton leak after **1[C,NH]** treatment is the result of increased ROS production. Remarkably, we found that **1[C,NH]** treated cells have well preserved cytoplasmic ATP production thus restricting **1[C,NH]** actions to the mitochondria.

Chemotherapy treatments focus on targeting specific mutations lead to resistance. However, by the moment, any cancer chemoresistance mechanism can overcome mitochondrial dysfunction. Nonetheless, cancers cannot evade a treatment that targets mitochondria redox homeostasis, which is essential for tumor progression. Here we describe small atomic modifications on the structure of the eight benzazole-based iridium(III) complexes relevant for novel pharmacological applications where redox state is important, like in cancer.

## Figures and Tables

**Figure 1 cancers-15-00107-f001:**
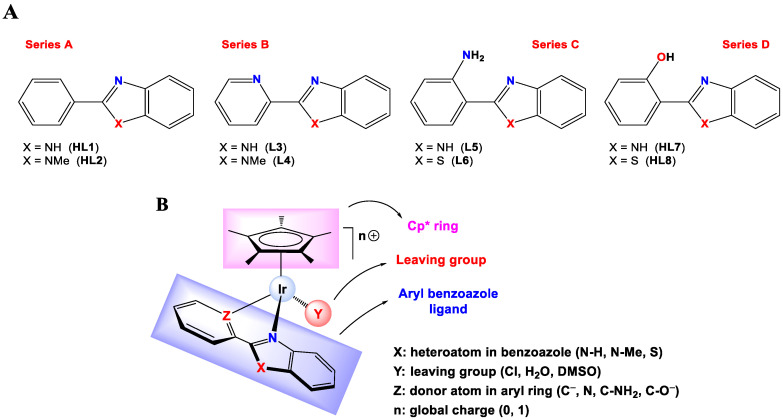
(**A**) Arylbenzazole ligands (L3, L4, L5 and L6) and proligands (HL1, HL2, HL7 and HL8) used in this work. (**B**) Schematic molecular structure of complexes studied in this work showing variable atoms.

**Figure 2 cancers-15-00107-f002:**
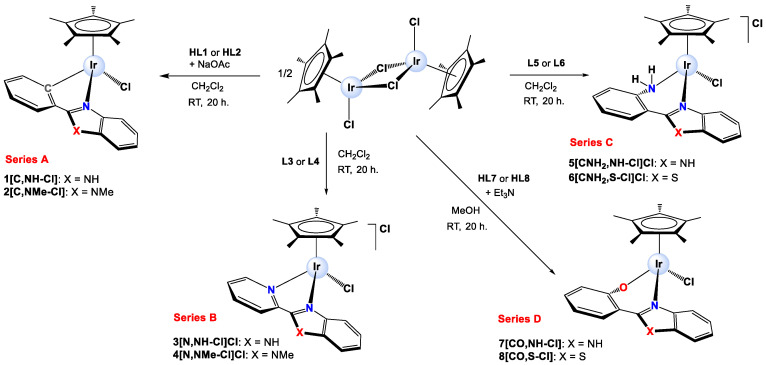
Schematic synthesis of the studied half-sandwich Iridium(III) complexes.

**Figure 3 cancers-15-00107-f003:**
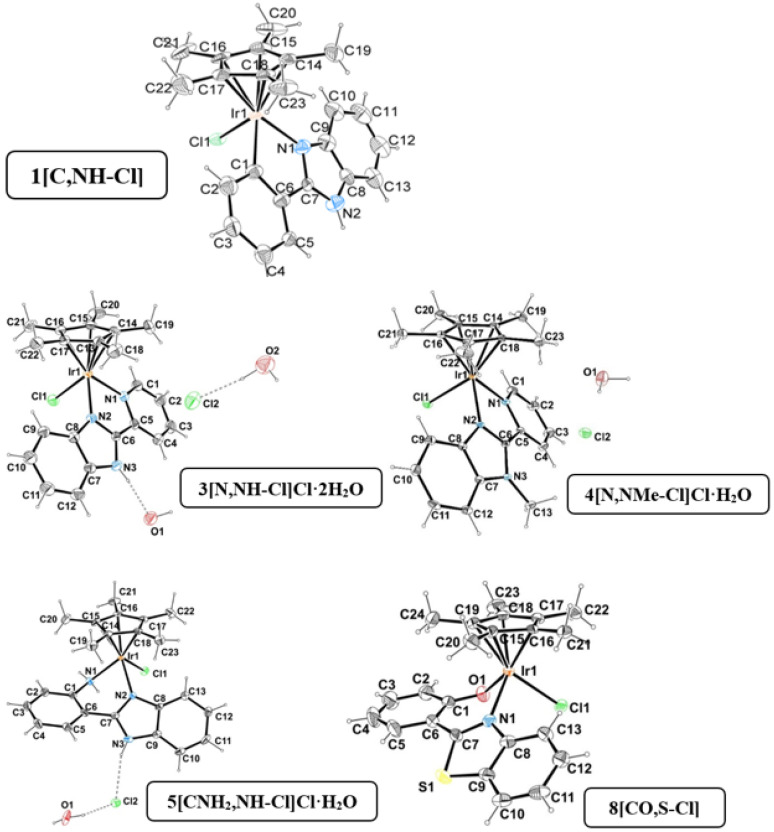
ORTEP diagrams for complexes **1[C,NH-Cl]**, **3[N,NH-Cl]·2H_2_O**, **4[N,Nme-Cl]·H_2_O**, **5[CNH_2_,NH-Cl]·H_2_O** and **8[CO,S-Cl]**. Ellipsoids are shown at 30% probability.

**Figure 4 cancers-15-00107-f004:**
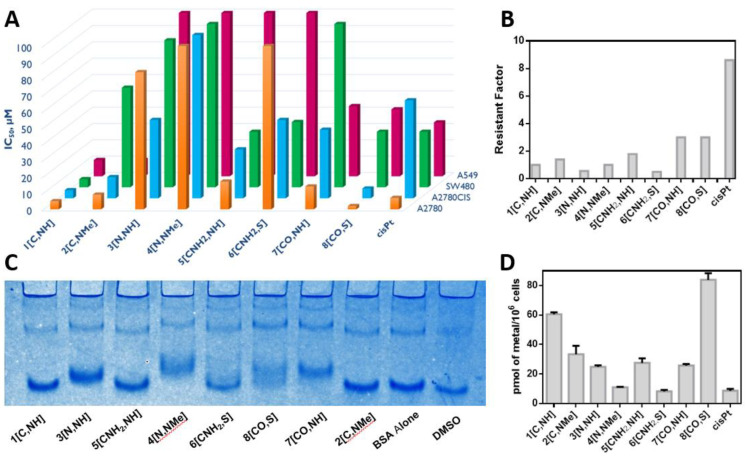
(**A**) Cytotoxic activity (IC_50_ expressed in µM) of iridium complexes at 48 h as in ovarian A2780, A2780cis (ovarian acquired resistance to cisplatin), colon SW480 and lung A549 cancer cells. (**B**) Ratio of the IC_50_ values of the studied complexes in A2780cis respecting those in A2780. (**C**) Native acrylamide gel electrophoresis of BSA in the presence of the iridium complexes, C_D_/C_P_=10. (**D**) Metal accumulation in SW480 cells treated during 24 h with 2 µM of the complexes. For original native PAGE gel image, see Appendix A.

**Figure 5 cancers-15-00107-f005:**
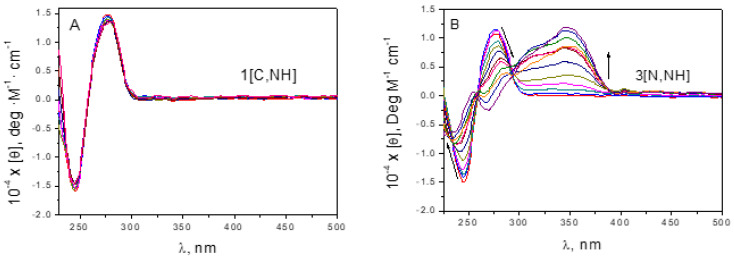
Circular dichroism spectra of DNA with (**A**) **1[C,NH]** and (**B**) **3[N,NH]**.

**Figure 6 cancers-15-00107-f006:**
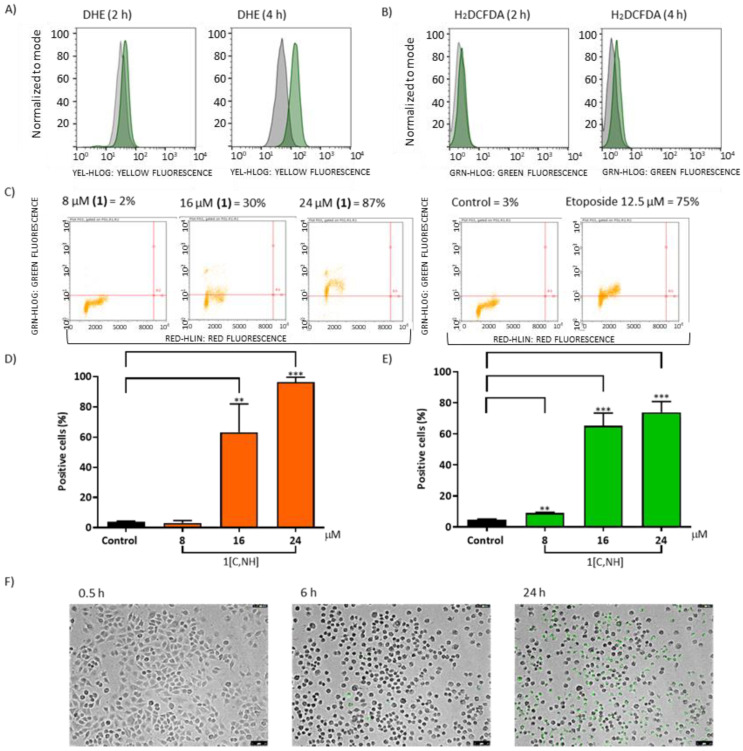
(**A**) O_2_^•-^ production after 2 h and 4 h incubation in A549 cells with **1[C,NH]** and detection using DHE by flow cytometry. (**B**) ROS production (H_2_O_2_ included) after 2 h and 4 h incubation in A549 cells with 16 µM of **1[C,NH]** and detection using H_2_DCFDA by flow cytometry. Green profile = **1[C,NH]**, Gray profile = Vehicle. 5000 stained cells were analyzed using the Guava EasyCyte flow cytometer and processed with the FlowJo program. Increases in fluorescence are indicative of increased ROS production. Images were representative of at least three independent experiments. (**C**) Percentage of A549 positive cells to damage on DNA analyzed by flow cytometry. Cells were treated for 4 h at the indicated doses of **1[C,NH]** (left). Untreated cells (control) and treated cells with Etoposide (positive control). (**D**) Cardiolipin oxidation. Percentage of positive cells. A549 cells were analyzed by flow cytometry after 4h incubation at the indicated concentrations of **1[C,NH]**. Data shown are the mean and SD of at least three independent experiments (** *p* < 0.01), *** *p* < 0.001; (**E**) Release Cytochrome c. Percentage of positive cells were measured using Millipore’s FlowCellect Cytochrome C Kit. A549 cells were incubated for 5 h at the indicated concentrations of **1[C,NH]** and analyzed by Flow Cytometry. Data shown are the mean and SD of at least three independent experiments (** *p* < 0.01, *** *p* < 0.001). (**F**) Caspase activity was determined using CellEvent Caspase 3/7 Green Detection Reagent. A549 cells were treated with 16 µM of **1[C,NH]** and observed using fluorescence microscope for 24 h. Images were captured at indicated incubation time. Green fluorescence emission indicates caspase activity (caspases 3/7). Magnification (10×).

**Figure 7 cancers-15-00107-f007:**
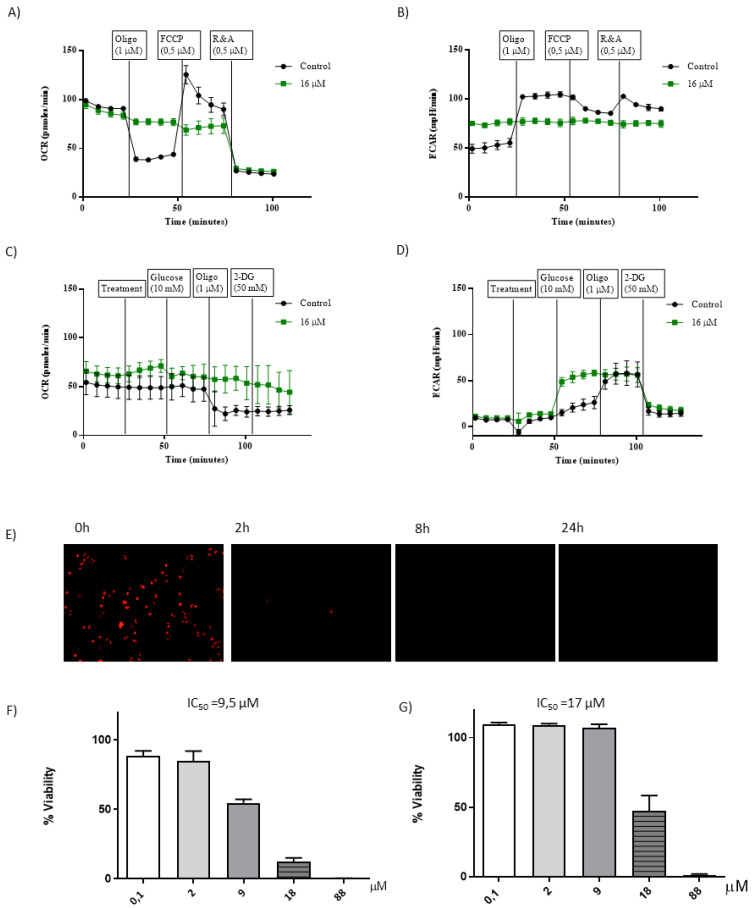
OCR (**A**) and ECAR (**B**) profile of A549 cells, incubated for 1 h with **1[C,NH]** (16 µM) or vehicle (control). (**C**) OCR and (**D**) ECAR profile of A549-gal incubated with (treatment) **1[C,NH]** (16 μM) or vehicle (control). Seahorse images are representative of three independent experiments. (**E**) A549 cells were pre-stained with TMRM (50 nM) and treated with 16 µM of **1[C,NH]**. Images (10×) were obtained at indicated incubation times after the treatment. Percentage of viability of A549 proliferant cells (**F**) and A549 quiescent cells (**G**), after 24 h of incubation at different concentrations of **1[C,NH]**. Data shown are the mean and SD of at least three independent experiments.

**Figure 8 cancers-15-00107-f008:**
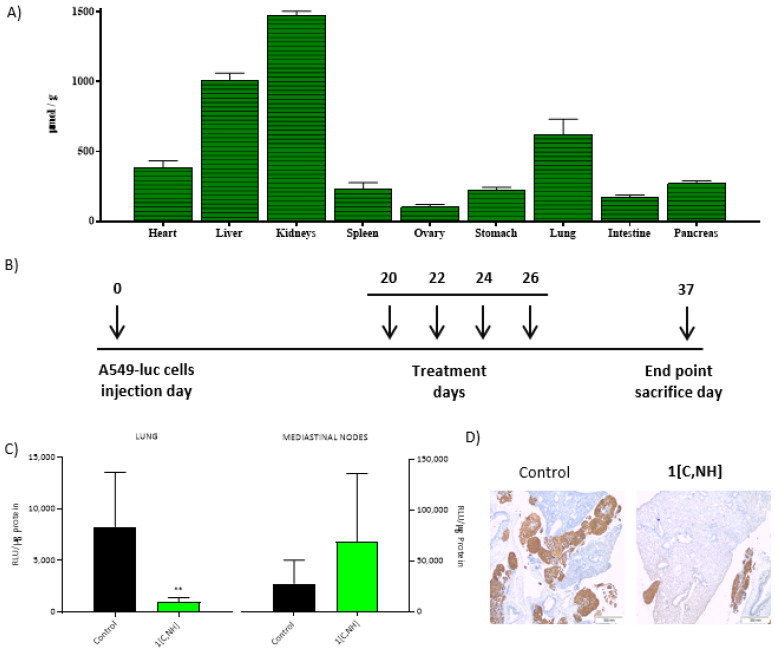
(**A**) Iridium biodistribution in extracted organs 4 h after i.v. injection of **1[C,NH]**. (**B**) In vivo monitoring scheme. (**C**) Luciferase activity quantification measured in protein extracts of lung or mediastinal nodes, ** *p* < 0.01. (**D**) Immunohistochemical staining of mouse lungs using anti-CK7 antibody. Left: control group. Right: Treated group. Brown stain shows tumor cells.

## Data Availability

The data presented in this study is available in this article (and Appendix A).

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
