# Peer review of "Targets, Mechanisms and Cytotoxicity of Half-Sandwich Ir(III) Complexes Are Modulated by Structural Modifications on the Benzazole Ancillary Ligand"

_cancers, 2022, doi:10.3390/cancers15010107_

Round 1

Reviewer 2 Report

Dear Authors, 

it was very interesting to review the presented manuscript demonstrating tremendous research efforts. 
The review summary and questions are given in the attached file.

Best regards.

Round 2

Reviewer 2 Report

Dear Authors,

Thank you very much for the responses to the previous questions and the edits in the revised manuscript. It is very much appreciated to see that all questions and comments have been addressed.

Some minor questions remain:

The explanation given as of line 984 refers to an article (reference 107), but not to actual data from the own study. Also other factors influence the efficacy of a drug on the primary tumor and not on the metastasis. The authors may reconsider and add potential other reasons, why tumor cells in the mediastinal nodes are less sensitive (biodistribution, cell-cell interactions, matrix in the nodes, etc.).

What is the reason of having used Balb/c mice for the bio distribution and nude mice for the orthotopic lung cancer model? Nude mice may have been used for both. It would be appreciated if the authors may give a statement in the manuscript.

The graph of the tumor growth in the response letter represents the tumor growth as % tumor growth and an increase of approx. 5000 % can be seen by day 33. Does the % value represent the total fluorescence measure without a further discrimination of the primary tumor and the metastatic tumor? Would the authors mind to show an IVIS image to demonstrate the way this has been calculated? In addition, may the authors explain what happens from day 28 to day 35? The error bars indicate that some tumors disappeared while others increased by a 2-fold.

In the second graph showing the weight of the mice, it appears that mice gain weight at the end of experiment. Does the weight gain (day 33-35) result from the tumor?

What is the actual tumor size and weight after dissection?

It would be appreciated if the authors may add the statement of the necropsy analysis in the manuscript.

My apologies for having it phrased not accurately enough. It was not meant to measure mitochondrial activity in vivo, but perform certain measures after the termination when the tissues were extracted. Would it be possible for the authors to still perform certain analysis (i.e. visualization of mitochondria) in the tissues that were removed from the euthanized animals.
